# DSCS: Fast CPDAG-Based Verification of Collapsible Submodels in High-Dimensional Bayesian Networks

**Wentao Wu[1], Shiyuan He[2], Jianhua Guo[2]***
[1]Northeast Normal University, [2]Beijing Technology and Business University

## Abstract

Bayesian networks (BNs), represented by directed acyclic graphs (DAGs), provide a principled framework for modeling complex dependencies among random variables. As data dimensionality increases into the tens of thousands, fitting and marginalizing a full BN becomes computationally prohibitive—particularly when inference is only needed for a small subset of variables. Estimation-collapsibility addresses this challenge by ensuring that directly fitting a submodel, obtained by ignoring non-essential variables, still yields exact inference on target variables. However, current DAG-based criterion for checking estimation-collapsibility is computationally intensive, involving exhaustive vertex searches and iterative removals. Additionally, practical applications typically identify the underlying DAG only up to its Markov equivalence class, represented by a completed partially directed acyclic graph (CPDAG). To bridge this gap, we introduce sequential $c$-simplicial sets—a novel graphical characterization of estimation-collapsibility directly applicable to CPDAGs. We further propose DSCS, a computationally efficient algorithm for verifying estimation-collapsibility within CPDAG framework that scales effectively to high-dimensional BNs. Extensive numerical experiments demonstrate the practicality, scalability, and efficiency of our proposed approach.

## 1 Introduction

*Bayesian networks* (BNs), represented by *directed acyclic graphs* (DAGs), offer a powerful and transparent framework to model complex dependency structures among random variables. By representing each variable as a vertex and each direct influence as a directed edge, BNs allow practitioners to read off causal or correlational relationships from their DAGs and to leverage powerful algorithms (such as message-passing) to compute marginals and posteriors efficiently. As a result, BNs have become indispensable tools across numerous domains, including econometrics [1], engineering [2–4], bioinformatics [5, 6], causal inference [7–9], and machine learning [10–12]. Recently, hybrid architectures that integrate DAG structure with deep neural networks have further extended BNs' expressive power while preserving their causal semantics [e.g. 13–15].

Despite these strengths, the explosive increase of data dimensionality has highlighted a critical bottleneck: constructing and marginalizing a full BN becomes infeasible when thousands—or even tens of thousands—of variables are involved, yet in many applications one cares about inference on only a small subset of variables. For instance, the Cancer Network Galaxy (TCNG) database hosts 768 networks encompassing over 20,000 genes and more than 16 million interactions. Typically, researchers only focus on disease-related or hub genes (genes with many interactions), which usually constitute a significantly smaller subset of the entire network. To draw inference and probability query for the target variables, it is both computationally expensive and wasteful to fit the full model and then marginalize to obtain a target submodel.

---

*Corresponding author: jhguo@btbu.edu.cn

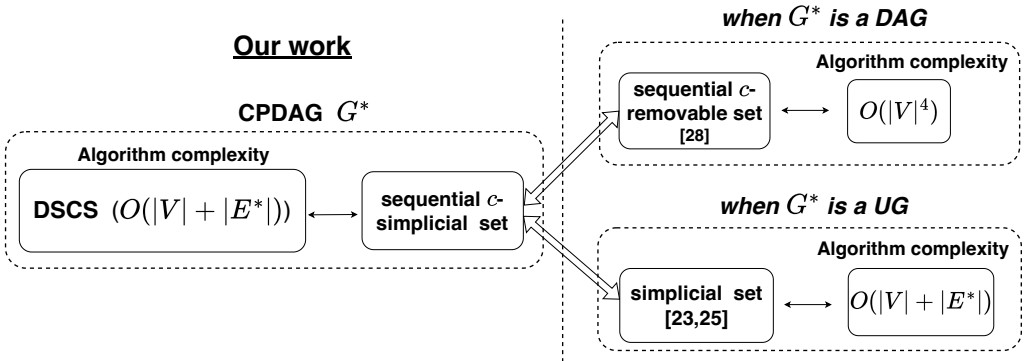

Figure 1: Criteria for verifying estimation-collapsibility along with their worst-case computational complexities. Our proposed CPDAG-based criterion, *sequential c-simplicial sets*, generalizes and unifies previous criteria for DAGs [28] and undirected graphs (UGs) [23, 25]. Our algorithm, DSCS, achieves a worst-case complexity of $O(|V| + |E^*|)$.

In such scenarios, we would like to "collapse" away irrelevant graph vertices (variables) and *directly* obtain an exact estimate of the submodel. This desideratum is captured by the notion of *estimation-collapsibility*, which ensures a submodel's direct estimation and inference aligns perfectly with that from the full-model, yielding drastic savings in data collection, computation, and robustness to unobserved variables [16–20]. However, neglecting proper validation of the submodel's collapsibility risks misleading or contradictory conclusions, exemplified by phenomena such as the Yule-Simpson paradox [21, 22]. Hence, identifying the precise conditions under which collapsibility holds is pivotal for trustworthy statistical inference with submodels.

Most existing works addressing collapsibility focused on undirected graphical models [e.g., 23–26], while the literature for Bayesian networks (or DAG models) remains notably limited [27, 28]. Despite its theoretical and practical significance, estimation-collapsibility in Bayesian networks remains as substantial challenges that are far from fully resolved. In particular, existing DAG-based criterion, *sequential c-removability* [28], can be computationally intensive to verify. It requires multiple laborious rounds of vertex examination, repeatedly identifying and removing c-removable vertices until none remain. The complexity can escalate up to $O(|V|^4)$, making the algorithm impractical for a high-dimensional Bayesian network with a large cardinality $|V|$ for its vertex set $V$.

Moreover, existing works [27, 28] assume the DAG structure is known, which is rarely true in practical scenarios. Typically, the underlying DAG must be inferred from observational data using causal discovery methods [29–31, 8, 32–36]. Lacking strong assumptions like linearity [33] or additive noise [31, 37], causal discovery process usually yields a Markov equivalence class, which is succinctly represented by a *completed partially directed acyclic graph* (CPDAG) [29, 38–40, 9, 41–44]. Consequently, a pressing open problem emerges:

*How can we efficiently verify estimation-collapsibility directly on the CPDAG for a high-dimensional Bayesian network?*

To address this challenge, the current work presents the following novel contributions:

- In Section 3.1, we introduce a novel concept, *sequential c-simplicial sets*, and find it as a necessary-and-sufficient criterion for estimation-collapsibility based on CPDAG.
- As shown in Figure 1, our framework recovers and unifies prior collapsibility results for DAGs and undirected graphs (UGs). When a CPDAG $G^*$ only has directed edges, a sequential c-simplicial set reduces to a sequential c-removable set of [28]. When $G^*$ only contains undirected edges, our sequential c-simplicial criterion is equivalent to the simplicial criterion for undirected graphs [23, 25].
- In Section 3.2, we develop the Detecting Sequential $C$-simplicial Sets (DSCS) algorithm, for rapid verification of estimation-collapsibility. It achieves a complexity order $O(|V| + |E^*|)$, where $|V|$ and $|E^*|$ denote the number of vertices and edges in a CPDAG $G^*$, respectively.

- Extensive experiments have been conducted in Section 4 to demonstrate our method's practical effectiveness and computational efficiency.

Lastly, we emphasize that while our method addresses typical scenarios where the underlying DAG is unknown, the DSCS Algorithm also offers significant advantages when the DAG is explicitly available. Indeed, verifying estimation-collapsibility is substantially simpler with our CPDAG-based criterion. Given a known DAG $\vec{G}$, converting it into its CPDAG representation $G^*$ and applying our DSCS Algorithm enables efficient verification at significantly reduced computational cost.

## 2 Preliminaries and Related Works

We begin by providing a review of graphical model concepts in Section 2.1. More terminologies and notations can be found in Appendix A. The notion of estimation-collapsibility for Bayesian networks is formally defined in Section 2.2. In Section 2.3, we review the existing criteria for checking estimation-collapsibility, as summarized in the right panel of Figure 1.

### 2.1 Graphical terminologies

A graph $\mathcal{G} = (V, E)$ is defined by a set $V$ of nodes (or vertices), and a set $E$ consisting of directed and (or) undirected edges. We use $\mathbf{pa}_{\mathcal{G}}(A)$, $\mathbf{ch}_{\mathcal{G}}(A)$, $\mathbf{ne}_{\mathcal{G}}(A)$, $\mathbf{an}_{\mathcal{G}}(A)$, $\mathbf{de}_{\mathcal{G}}(A)$, and $\mathbf{mb}_{\mathcal{G}}(A)$ to denote the union of the parents, children, neighbors, ancestors, descendants, and the Markov boundary of each vertex in a set $A \subseteq V$ in $\mathcal{G}$, respectively, and with the set $A$ excluded after taking the set union. Note that, among these, $\mathbf{ne}_{\mathcal{G}}(A)$ is defined with respect to undirected edges, while the other five notions are defined in terms of directed edges. Formal definitions can be found in Appendix A. If $A$ needs to be included, we simply capitalize the first letter of the corresponding symbol, e.g., $\mathbf{Pa}_{\mathcal{G}}(A) \triangleq \mathbf{pa}_{\mathcal{G}}(A) \cup A$.

**UGs, DAGs, CPDAGs.** If $\mathcal{G}$ consists solely of undirected edges, it is referred to as an *undirected graph* (UG), denoted as $G = (V, E)$. When $\mathcal{G}$ is composed entirely of directed edges and contains no directed cycles, it is referred to as a *directed acyclic graph* (DAG), denoted as $\vec{G} = (V, \vec{E})$. A DAG encodes a set of conditional independence relations based on the notion of *d-separation* [11]. Two DAGs are Markov equivalent if they encode the same set of conditional independence relations. A Markov equivalence class of a DAG $\vec{G}$ can be uniquely represented by a *completed partially directed acyclic graph* (CPDAG) [45], denoted by $G^* = (V, E^*)$. The undirected components of $G^*$ are *undirected and connected chordal graphs* (UCCGs) [46], also known as *chain components* of $G^*$ [7, 46]. We use $\mathcal{M}(G^*)$ to denote the set of all Markov equivalent DAGs represented by the CPDAG $G^*$.

### 2.2 Bayesian network and estimation-collapsibility

For a DAG $\vec{G} = (V, \vec{E})$, suppose $X_V$ is a random vector indexed by the graph vertices. For $A(\subseteq V)$, let $X_A$ be the sub-vector indexed by the subset $A$. A joint distribution $P$ over $X_V$ is *compatible* with $\vec{G}$ if it holds that $P(x_V) = \prod_{v \in V} P(x_v \mid x_{\mathbf{pa}_{\vec{G}}(v)})$ for any $x_V$. This means that $P$ factorizes according to the parents of each vertex $v$ in $\vec{G}$. The family of distributions compatible with $\vec{G}$ is denoted as $\mathcal{P}(\vec{G})$ and the pair $\mathcal{B} = (\vec{G}, \mathcal{P}(\vec{G}))$ is termed a *Bayesian network* (BN) or *DAG model*.

For a BN, estimation-collapsibility can be formally defined via the commutativity of model fitting and marginalization [47]. Let $R(\subseteq V)$ be the subset of nodes of interests, and let $M \triangleq V \setminus R$ represent the non-irrelevant variables we want to marginalize out. Denote $\hat{P}(x_R)$ as the estimate of $P(x_R)$ obtained by marginalizing the maximum likelihood estimate (MLE) $\hat{P}(x)$ under the full DAG model $(\vec{G}, \mathcal{P}(\vec{G}))$. Meanwhile, let $(\vec{G}_R, \mathcal{P}(\vec{G}_R))$ be the induced model, where the graph $\vec{G}_R$ is obtained by removing the vertex set $M$ and all edges incident to any vertex in $M$. We let $\hat{P}_{\vec{G}_R}(x_R)$ be the MLE of the joint probability under the induced model $(\vec{G}_R, \mathcal{P}(\vec{G}_R))$. Estimation-collapsibility means that the MLE obtained from the original DAG model, after marginalization, equals the MLE from the induced DAG model.

**Definition 1.** *[see 27, Definition 1] Suppose $\mathcal{B} = (\vec{G}, \mathcal{P}(\vec{G}))$ is a Bayesian network, $\mathcal{P}(\vec{G})$ is estimation-collapsible over a single vertex $v$ (or onto $V \setminus \{v\}$ ) if for any $x_{V \setminus \{v\}}$, it holds that*

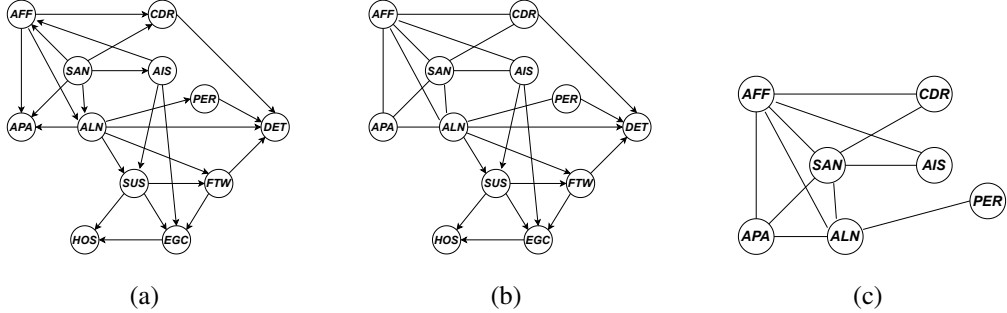

(a)          (b)          (c)

Figure 2: *A DAG from [49] depicting the assumed causal relations between 12 prodromal symptoms of schizophrenia. Fig. 2(a) shows the causal DAG, and Fig. 2(b) displays the corresponding CPDAG. Fig. 2(c) presents the largest chain component in the CPDAG. The nodes represent:* AFF= *Affective Flattening,* AIS= *Active Isolation,* ALN = *Alienation,* APA= *Apathy,* CDR= *Cognitive Derailment,* DET= *Delusional Thinking,* EGC= *Egocentrism,* FTW= *Living in a Fantasy World,* HOS= *Hostility,* PER= *Perceptual Aberrations,* SAN= *Social Anxiety,* SUS= *Suspiciousness.*

$\hat{P}(x_{V \setminus \{v\}}) = \hat{P}_{\vec{G}_{V \setminus \{v\}}}(x_{V \setminus \{v\}})$. *Estimation-collapsibility over a set $M$ (or onto $R \triangleq V \setminus M$ ) holds if $\hat{P}(x_R) = \hat{P}_{\vec{G}_R}(x_R)$.*

When this estimation-collapsibility property holds, we only need to work with the induced sub-model $(\vec{G}_R, \mathcal{P}(\vec{G}_R))$ to directly obtain the estimate $\hat{P}(x_R)$. The obvious benefit is a substantial saving of data collection and computational efforts while maintaining exact estimate and inference for a high-dimensional Bayesian network.

### 2.3 Existing criteria for checking estimation-collapsibility

**2.3.1 Criterion for DAG**. Kim & Kim [27] discussed estimate-collapsibility for directed acyclic graph (DAG) models of contingency tables [48]. Xie & Geng [28] further investigated for discrete and continuous variable DAG models, and introduced the important concepts below.

**Definition 2.** *[see 28, Definition 3] A vertex $v$ is $c$-removable from DAG $\vec{G}$ if any two vertices in $\mathbf{Mb}_{\vec{G}}(v) \triangleq \{v\} \cup \mathbf{mb}_{\vec{G}}(v)$ are adjacent, except when both vertices belong to $\mathbf{pa}_{\vec{G}}(v)$. A set $M$ is sequentially $c$-removable if all vertices in $M$ can be ordered such that they can be $c$-removed by that ordering.*

Furthermore, relying on the mild assumption of model non-triviality [see 28, Definition 4]—a condition satisfied by most DAG models, including those for contingency tables and Gaussian distributions [28] and implicitly used by [27]—they established the equivalence between *sequential c-removability* and estimation-collapsibility for DAG models.

**Proposition 1.** *[see 28, Theorem 2] Suppose that $\mathcal{B} = (\vec{G}, \mathcal{P}(\vec{G}))$ is a non-trivial DAG model, then $M$ is sequentially $c$-removable from $\vec{G}$ if and only if $\mathcal{P}(\vec{G})$ is estimate-collapsible over $M$.*

The above theorem enables us to check estimation-collapsibility via examining the sequential $c$-removability. The next example illustrates this procedure.

**Example 1.** *Consider the DAG in Figure 2(a). Suppose we are interested in finding whether the model is estimate collapsible over $M = \{\text{AFF}, \text{APA}, \text{SAN}\}$. Checking sequential c-removability requires multiple rounds of iterations. In the first round, we find that neither* AFF *nor* SAN *is c-removable from $\vec{G}$, but* APA *is c-removable. After removing* APA, *the second round is repeated by examining if any node in $\{\text{AFF}, \text{SAN}\}$ is c-removable from $\vec{G}_{V \setminus \{\text{APA}\}}$. Upon checking, we find that no vertex in $\{\text{AFF}, \text{SAN}\}$ is further c-removable. The procedure stops by concluding that the model is not estimate collapsible over $M$.*

The above procedure quickly becomes infeasible as the graph size grows. Given a subset $M$ with cardinality $|M|$, it requires $O(|M|)$ iterations of meticulous vertex search. In each round, one must

scan the remaining vertices in $M$ to identify the $c$-removable ones and then remove them. In the worst case—when the graph is dense and $|M|$ is on the same order as $|V|$—the first iteration alone costs $O(|V|^3)$, the next $O((|V|-1)^3)$, and so on, leading to a total complexity on the order of $O(|V|^4)$.

**2.3.2 Criterion for UG**. As our criterion also recover those for undirected graph (UG), we conclude this section by briefly reviewing some important results.

**Definition 3.** *[see 50] In an undirected graph (UG) $G = (V, E)$, for a vertex $v \in V$, if the neighborhood $\mathbf{ne}_G(v)$ induces a complete subgraph in $G$, then $v$ is called a simplicial vertex in $G$. For a subset $M \subseteq V$, if the neighborhood $\mathbf{ne}_G(M)$ induces a complete subgraph in $G$, then $M$ is called a simplicial (vertex) set in $G$.*

For multinomial and Gaussian undirected graphical models, the collapsibility over $M$ is equivalent to the condition that each connected component of $M$ is a simplicial set [23, 25]. The complexity of checking this simplicial property is $O(|V| + |E|)$.

# 3 Fast CPDAG-Based Verification of Estimation-Collapsibility

We now address the main query of the current work: *how can we efficiently check estimation-collapsibility for high-dimensional Bayesian Networks using CPDAGs?* In Section 3.1, we will establish an equivalent characterization of estimation-collapsibility from the perspective of CPDAGs. In Section 3.2, we develop an efficient and scalable algorithm to check the estimation-collapsibility.

## 3.1 Sequential $c$-simplicial set

For a Bayesian network (BN), checking estimation-collapsibility of the underlying DAG via the corresponding CPDAG $G^*$ is possible, as this property is consistent across all DAGs within the equivalence class $\mathcal{M}(G^*)$. In fact, for two Markov-equivalent DAGs $\vec{G}_1$ and $\vec{G}_2$ in $\mathcal{M}(G^*)$, they encode the same set of conditional independence relations, ensuring that the families of distributions they encode are identical, i.e., $\mathcal{P}(\vec{G}_1) = \mathcal{P}(\vec{G}_2)$. Similarly, for Markov-equivalent subgraphs, the corresponding families of distributions are also identical. Therefore, when estimation-collapsibility holds for the underlying DAG, the property is preserved across all its Markov equivalent DAGs.

Building on this consistency, we re-state estimation-collapsibility (see Definition 1) from the perspective of CPDAG. The definition below formalizes this extension based on the family of distributions compatible with a CPDAG. For a CPDAG $G^* = (V, E^*)$, a joint distribution $P$ over $X_V$ is said to be compatible with $G^*$ if it is compatible with a DAG $\vec{G}$ in $\mathcal{M}(G^*)$. The set of all such distributions is denoted as $\mathcal{P}(G^*)$.

**Definition 4.** *For a CPDAG $G^* = (V, E^*)$ and some $M \subseteq V$, $\mathcal{P}(G^*)$ is estimation-collapsible over the set $M$ if for every DAG $\vec{G} \in \mathcal{M}(G^*)$, $\mathcal{P}(\vec{G})(= \mathcal{P}(G^*))$ is estimation-collapsible over the set $M$.*

It is evident that, to examine the estimation-collapsibility for the underlying DAG model, we only need to check the estimation-collapsibility from CPDAG. Analogous to sequential $c$-removability in DAGs, we introduce the concept of a sequential $c$-simplicial (vertex) set for CPDAGs, which proves to be a key for addressing the problem.

**Definition 5.** *Given a CPDAG $G^* = (V, E^*)$, for a vertex $v \in V$, if $\mathbf{ch}_{G^*}(v) = \emptyset$ and $\mathbf{ne}_{G^*}(v)$ is complete, then $v$ is called a $c$-simplicial vertex in $G^*$. Similarly, for $M \subseteq V$, if every vertex in $M$ can be identified as a $c$-simplicial vertex during successive removals according to some vertex order, then $M$ is called a sequential $c$-simplicial (vertex) set in $G^*$.*

**Remark 1.** *The set $\mathbf{ch}_{G^*}(v)$ consists of nodes pointed to by directed edges from node $v$, i.e., $\mathbf{ch}_{G^*}(v) = \{w \in V \setminus \{v\} : v \to w \text{ in } G^*\}$, while $\mathbf{ne}_{G^*}(v)$ refers to the set of nodes adjacent to $v$ via undirected edges, i.e., $\mathbf{ne}_{G^*}(v) = \{w \in V \setminus \{v\} : v - w \text{ in } G^*\}$.*

**Example 2.** *Consider the CPDAG in Figure 2(b). Since $\mathbf{ch}_{G^*}(\text{APA}) = \emptyset$ and its neighbor set $\mathbf{ne}_{G^*}(\text{APA}) = \{\text{AFF, ALN, SAN}\}$ is complete, the node APA is a $c$-simplicial vertex in $G^*$. Similarly, HOS is also a $c$-simplicial vertex in $G^*$. The subset of nodes $M \triangleq \{\text{AIS, SUS, FTW, DET, HOS, EGC}\}$ forms a sequential $c$-simplicial vertex set in $G^*$, as $M$ can be identified as a $c$-simplicial vertex during successive removals according to the vertex order DET, HOS, EGC, FTW, SUS, AIS.*

The concept of a *c-simplicial vertex* in a CPDAG generalizes the notion of a *leaf node* (i.e., a node with an empty set of children) in DAGs and a *simplicial vertex* (i.e., a node whose set of neighbors is complete) [50] in undirected graphs. In particular, when $G^*$ is a DAG, $v$ is a *c-simplicial vertex* if and only if $v$ is a *leaf node*. This is because when $G^*$ is a DAG, it naturally holds that $\mathbf{ne}_{G^*}(v) = \emptyset$, making the completeness of its neighbor trivially satisfied, and the only condition required is $\mathbf{ch}_{G^*}(v) = \emptyset$. Meanwhile, when $G^*$ is an undirected graph, $v$ being a *c-simplicial vertex* in the CPDAG is equivalent to $v$ being a *simplicial vertex* in the undirected graph $G^*$. In this case, $\mathbf{ch}_{G^*}(v) = \emptyset$ always holds, and the sole requirement is that $\mathbf{ne}_{G^*}(v)$ forms a complete subgraph. The "$c$" in $c$-simplicial stands for *compound*, as the $c$-simplicial property (CPDAG) combines the features of a simplicial set (undirected graph) and a leaf set (DAG).

The notion of sequential $c$-simplicial set likewise generalizes sequential $c$-removable set for DAG. When $G^*$ is a DAG and $M$ is a sequential $c$-simplicial set, it directly follows from the previous discussion that $M$ is also sequentially $c$-removable.

Furthermore, we can find the concept of sequential $c$-simplicial set generalizes that of simplicial set for undirected graph. When $G^*$ contains only undirected edges (i.e. $G^*$ is a chordal graph), recall a $c$-simplicial vertex reduces to a simplicial vertex. This means that, for a sequential $c$-simplicial $M$, its vertices can be ordered such that each vertex is a simplicial vertex at the time of its removal. Consequently, in a chordal graph $G$, we also term a sequential $c$-simplicial set $M$ as a *sequential simplicial set*. Lemma 1 indicates that sequential simplicity (or sequential $c$-simplicity) is equivalent to simplicity for chordal graph.

**Lemma 1.** *For a chordal graph $G = (V, E)$, given a connected subset $M \subseteq V$, $M$ forms a sequential simplicial set in $G$ if and only if $M$ is a simplicial set in $G$.*

**Remark 2.** *In a chordal graph $G$, if $M$ is disconnected, its connected components can be analyzed independently. Then, $M$ is a sequential simplicial set in $G$ if and only if each connected component of $M$ is a simplicial set in $G$.*

**Remark 3.** *Lemma 1 connects the classical concept of simplicial set [50] and our concept of sequential simplicial set for a chordal graph. The correctness of Lemma 1 follows directly from Proposition B.1 in Appendix B. For general undirected graphs, the relationship between the two concepts is also discussed in detail in Appendix B. Table C.1 in Appendix C consolidates the key concepts of simplicial, $c$-removable, and $c$-simplicial vertices and sets across UGs, DAGs, and CPDAGs to facilitate comparison and reference.*

**Example 3.** *Consider Figure 2(c), which we denote as $G^*_{\tau_{max}}$. Clearly, $G^*_{\tau_{max}}$ is a chordal graph. Now, consider the subset $M \triangleq \{\text{AIS}, \text{CDR}, \text{SAN}\}$. $M$ is a simplicial set in $G^*_{\tau_{max}}$ because $\mathbf{ne}_{G^*_{\tau_{max}}}(M) = \{\text{AFF}, \text{APA}, \text{ALN}\}$ is complete. Additionally, $M$ forms a sequential simplicial vertex set in $G^*_{\tau_{max}}$, as $M$ can be identified as a simplicial vertex during successive removals according to the vertex order CDR, AIS, SAN.*

The preceding discussion underscores the profound connection between estimation-collapsibility and our notion of sequential $c$-simplicity in two classical situations. When the CPDAG is a DAG, if $M$ is sequential $c$-simplicial (hence sequential $c$-removable), then $\mathcal{P}(G^*)$ is estimate-collapsible over the set $M$ under the non-trivial model assumption [28, Definition 4]. Meanwhile, when the CPDAG is a undirected chordal graph, if $M$ is sequential $c$-simplicial (hence each componnent is simplicial), then the estimate-collapsiblility also holds in multinomial and Gaussian undirected graphical models [23, 25]. These relations are summarized in Figure 1 and lead us to the question: beyond these two special cases, can we establish a connection between estimation-collapsibility and sequential $c$-simplicity for a CPDAG in general? Theorem 1 below provides an affirmative answer.

**Theorem 1.** *Given a CPDAG $G^* = (V, E^*)$, let $\mathcal{M}(G^*)$ denote the set of all Markov equivalent DAGs represented by the CPDAG $G^*$. For $M \subseteq V$, the following four statements are equivalent:*
*(1) $M$ is a sequential $c$-simplicial set in $G^*$;*
*(2) there exists a DAG $\vec{G} \in \mathcal{M}(G^*)$ such that $\mathbf{ch}_{\vec{G}}(M) = \emptyset$;*
*(3) for all DAGs $\vec{G} \in \mathcal{M}(G^*)$, $M$ is sequentially $c$-removable from $\vec{G}$;*
*(4) under the non-trivial model assumption, $\mathcal{P}(G^*)$ is estimate-collapsible over the set $M$.*

Theorem 1 establishes the connection between estimation-collapsibility and sequential $c$-simplicial sets. It clearly shows that our notion of a $c$-simplicial set naturally extends to the broader framework of CPDAG models, including DAGs and undirected chordal graphs as special cases. Specifically,

under the non-trivial model assumption [28, Definition 4], it can be employed to verify estimation-collapsibility for a high-dimensional Bayesian network. The proof of Theorem 1 is in Appendix D.

## 3.2 DSCS: fast CPDAG-based verification of estimation-collapsibility

Although Theorem 1 establishes CPDAG-based criterion for checking estimation-collapsibility, directly determining if a subset $M$ is a sequential $c$-simplicial remains challenging. This is because multiple iterations are still needed to search for the $c$-simplicial vertices, which is a daunting task. Fortunately, for a CPDAG, we can find an equivalent characterization of sequential $c$-simplicity, which does not involve vertex order and eliminates the requirement of multiple algorithm iterations.

**Remark 4.** *To highlight the essential concepts and intuitions, the following discussion focuses on the case where $M$ is connected. For a disconnected $M$, the conclusion can be easily generalized as each connected component can be considered separately. In the Appendix, the theoretical derivations are developed for the general case where $M$ has several connected components.*

First, we establish a necessary condition for $M$ to be a sequential $c$-simplicial set.

**Lemma 2.** *Suppose $G^* = (V, E^*)$ is a CPDAG. For any $M \subseteq V$, if $M$ is a sequential $c$-simplicial set in $G^*$, then it holds that $\mathbf{ch}_{G^*}(M) = \emptyset$.*

If $\mathbf{ch}_{G^*}(M) \neq \emptyset$, then by Lemma 2, it follows directly that $M$ is not a sequential $c$-simplicial set in $G^*$, thereby avoiding further effort for verification. However, the condition $\mathbf{ch}_{G^*}(M) = \emptyset$ alone does is not sufficient to ensure that $M$ is a sequential $c$-simplicial set. Consider the CPDAG in Figure 2(b) for example, and let $M = \{\text{AFF, SAN}\}$. Although $\mathbf{ch}_{G^*}(M) = \emptyset$, neither AFF nor SAN is a $c$-simplicial vertex in $G^*$. It follows that $M$ is not a sequential $c$-simplicial set in $G^*$.

In addition to requiring $\mathbf{ch}_{G^*}(M) = \emptyset$, additional conditions must be imposed for sequential $c$-simplicity to hold. We start by examining a special case where $M$ is contained within a single chain component, and provide an equivalent characterization of its sequential $c$-simplicial property.

**Lemma 3.** *Suppose $G^* = (V, E^*)$ is a CPDAG with its chain components denoted by $\mathcal{T} = \{\tau_1, \tau_2, \ldots, \tau_K\}$. For a connected subset $M \subseteq V$, if there exists $i \in [K]$ such that $M \subseteq \tau_i$, then the following two statements are equivalent:*
  *(1) $M$ is a sequential $c$-simplicial set in $G^*$;*
  *(2) $\mathbf{ch}_{G^*}(M) = \emptyset$, and $M$ is a simplicial set in $G^*_{\tau_i}$, meaning that $\mathbf{ne}_{G^*_{\tau_i}}(M)$ forms a complete subgraph.*

**Remark 5.** *For a disconnected $M$, it is sufficient to impose the constraints separately on each of its connected components. In this case, statement (2) in Lemma 3 can be easily reformulated as follows:*

> *(2′) $\mathbf{ch}_{G^*}(M) = \emptyset$, and each connected component of $M$ is a simplicial set in the subgraph $G^*_{\tau_i}$.*

We continue to consider the general case where a connected $M$ is contained within multiple chain components of $G^*$. Let $\mathcal{T} = \{\tau_1, \tau_2, \ldots, \tau_K\}$ be the set of chain components of $G^*$, and define $\alpha$ as a mapping from $\mathcal{T}$ to $\{1, 2, \ldots, K\}$ such that if there is a directed edge from $\tau_i$ to $\tau_j$, then $\alpha(\tau_i) < \alpha(\tau_j)$. This mapping $\alpha$ is called a *topological ordering* of the chain components in $G^*$. Without loss of generality, assume $\alpha(\tau_1) < \alpha(\tau_2) < \cdots < \alpha(\tau_K)$. For $M$ to be a sequential $c$-simplicial set in $G^*$, in addition to requiring $\mathbf{ch}_{G^*}(M) = \emptyset$, it suffices that each $M_i \triangleq M \cap \tau_i$ is a sequential simplicial set in $G^*_{\tau_i}$ for all $i \in [K]$. By Lemma 1, this is equivalent to requiring each connected $M_i$ in $G^*_{\tau_i}$ forms a simplicial set. This is because we can always remove the sets in the reverse order $M_K, M_{K-1}, \cdots, M_1$. The specific conclusion is given by Theorem 2.

**Theorem 2.** *Given a CPDAG $G^* = (V, E^*)$ with its chain component set $\mathcal{T} = \{\tau_1, \tau_2, \ldots, \tau_K\}$, for connected $M \subseteq V$, let $M_i \triangleq M \cap \tau_i$ for each $i \in [K]$. The following two statements are equivalent:*
  *(1) $M$ is a sequential $c$-simplicial set in $G^*$;*
  *(2) $\mathbf{ch}_{G^*}(M) = \emptyset$, and $M_i$ in $G^*_{\tau_i}$ being a simplicial set for all $i \in [K]$.*

**Remark 6.** *For a disconnected $M$, statement (2) in Theorem 2 naturally extends to:*

> *(2′) $\mathbf{ch}_{G^*}(M) = \emptyset$, and each connected component of $M_i$ in $G^*_{\tau_i}$ being a simplicial set for all $i \in [K]$.*

---

**Algorithm 1:** DSCS Algorithm

---

**Input** : A CPDAG $G^* = (V, E^*)$, $M \subseteq V$.

**Output :** If $M$ is a sequential $c$-simplicial set in $G^*$, the algorithm returns TRUE; otherwise, the algorithm returns FALSE.

**1 if** $ch_{G^*}(M) \neq \emptyset$ **then**

**2**   **return** FALSE;

**3** Search for the set of chain components of $G^*$, and denote them as $\{\tau_1, \tau_2, \cdots, \tau_K\}$;

**4 for** $i \leftarrow 1$ **to** $K$ **do in parallel**

**5**   **if** $M \cap \tau_i \neq \emptyset$ **then**

**6**    Search for the connected components of $M \cap \tau_i$, and denote them as $\{M_1, M_2, \cdots, M_{l_i}\}$;

**7**    **for** $j \leftarrow 1$ **to** $l_i$ **do in parallel**

**8**     **if** $ne_{G^*_{\tau_i}}(M_j)$ *is not complete* **then**

**9**      **return** FALSE;

**10 return** TRUE.

---

The significance of Theorem 2 is that it provides an order-free way to check sequential $c$-simplicity. Recall from Section 3.1 that sequential $c$-simplicity for CPDAG implies estimation-collapsibility for the underlying DAG model. Therefore, Theorem 2 provides a convenient avenue to determine the estimation-collapsibility. The proof of Theorem 2 can be found in Appendix D.

Building on Theorem 2, we introduce the **D**etecting **S**equential **C**-simplicial **S**et (**DSCS**) algorithm in Algorithm 1. The algorithm proceeds in two steps. In the first step (Lines 1–2), we check whether $\mathbf{ch}_{G^*}(M)$ is empty; if not, we can immediately conclude that $M$ is not a sequential $c$-simplicial set in $G^*$. If $\mathbf{ch}_{G^*}(M)$ is empty, we proceed to the second step (Lines 3–19). For each chain component $\tau_i$ and each connected component $M_{l_j}$ of $M$ within $\tau_i$, we examine whether its neighboring nodes of $M_{l_j}$ form a complete subgraph. Note that these checks are independent across chain components in parallel. The next example illustrates the execution of the algorithm.

**Example 4.** *Let us re-examine Example 1 and recall we are interested in checking estimate-collapsibility with* $M = \{\text{AFF}, \text{APA}, \text{SAN}\}$. *Our DSCS algorithm works with the CPDAG in Figure 2(b). The CPDAG has chain components* $\tau_1 = \{\text{AFF}, \text{APA}, \text{ALN}, \text{SAN}, \text{PER}, \text{CDR}, \text{AIS}\}$, $\tau_2 = \{\text{SUS}\}$, $\tau_3 = \{\text{FTW}\}$, $\tau_4 = \{\text{EGC}\}$, $\tau_5 = \{\text{HOS}\}$, *and* $\tau_6 = \{\text{DET}\}$. *DSCS starts by checking if* $\mathbf{ch}_{G^*}(M) = \emptyset$, *which is clearly true. Next, it checks whether each connected component of* $M \cap \tau_i$ *in* $G^*_{\tau_i}$ *for* $i = 1, 2, \cdots, 6$ *is a simplicial set. From Figure 2(b), it is evident that* $M \subset \tau_1$, *so only the connected components of* $M$ *in* $G^*_{\tau_1}$ *need to be checked for being simplicial. Since* $M$ *is connected in* $G^*_{\tau_1}$, *and* $\mathbf{ne}_{G^*_{\tau_1}}(M) = \{\text{ALN}, \text{AIS}, \text{CDR}\}$ *is not complete, we conclude the underlying DAG model is not estimate-collapsible over* $M$.

The complexity of DSCS Algorithm is $O(|V| + |E^*|)$, where $|V|$ and $|E^*|$ denote the number of vertices and edges in $G^*$, respectively. While operations such as identifying the children subset $\mathbf{ch}_{G^*}(M)$, detecting connected components in $M$, and analyzing their neighborhoods have relatively low complexity, the chain component decomposition of $G^*$, requiring $O(|V| + |E^*|)$, constitutes the primary bottleneck. Following reviewer feedback, we can use the *maximum cardinality search* (MCS) algorithm [51–53] — which identifies all maximal cliques in a chordal graph and finds a perfect elimination ordering in linear time — to perform parallel verification of whether each neighborhood forms a clique across all chain components. This is done by checking whether the neighborhood is a subset of the maximal clique associated with the node (within that neighborhood) that appears earliest in the elimination ordering. Consequently, the complexity of Line 8 in the DSCS algorithm is $O(|\tau_{max}| + |E^*_{\tau_{max}}|)$, where $|\tau_{max}|$ is the number of vertices in the largest chain component $\tau_{\max}$, and $|E^*_{\tau_{max}}|$ is the number of edges in this component. Thus, the overall complexity of the DSCS algorithm is $O(|V| + |E^*|)$.

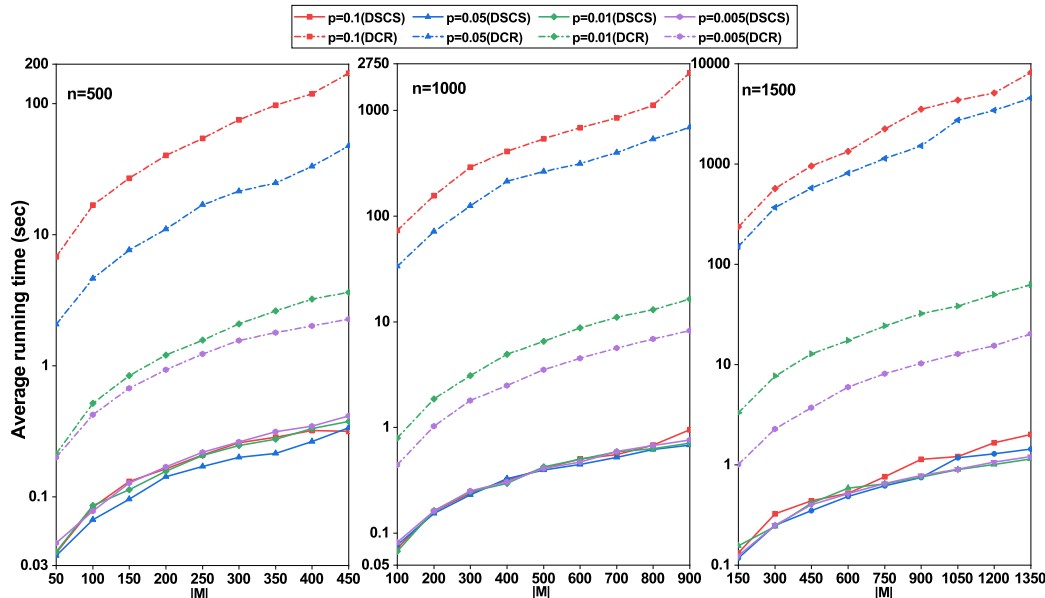

Figure 3: The average running time of DSCS (solid curves) versus DCR (dash-dotted curves) for different values of $n = |V|$, $p$, and $|M|$. The y-axis (average running time in seconds) is in logarithmic scale, and the x-axis represents $|M|$. The three panels correspond to different values of $n$.

## 4  Simulation

Through numerical experiments, we examine the performance of the proposed **DSCS** Algorithm in Algorithm 1. Our algorithm is compared to the approach of directly detecting sequential $c$-removability of $M$, as proposed by [28]. The latter approach is abbreviated as **DCR**. The experiments were implemented with $R$ and run on a computer with 2.20GHz CPU and 256 GB memory.

In the experiments, Erdös-Rényi graphs were randomly generated using the R-package pcalg [54], and the corresponding CPDAGs were obtained from the generated DAGs. We use $n = |V|$ to denote the number of graph vertices and $p$ to represent the edge creation probability. In each simulation replicate, we randomly generate a DAG with $n \in \{500, 1000, 1500, 2000, 4000, 6000, 8000, 10000\}$ and $p \in \{0.1, 0.05, 0.01, 0.005\}$, and the vertex subset $M$ was randomly selected from the graph with cardinality $|M|$. DCR (or DSCS) is then applied to the genrated DAGs (or corresponding CPDAGs) to check if the models are estimate-collapsible over $M$. For each combination of $n$, $p$, and $|M|$, the simulation was run 30 times. The average running time (in seconds) and the standard error were then calculated. Note that the proposed DSCS Algorithm can be run for each chain component of a CPDAG in parallel. In this experiment, only serial execution is implemented. Due to the extremely long computational time of the DCR Algorithm for large graphs, we only compared the average running time of DSCS with DCR for $n \in \{500, 1000, 1500\}$. When $n \in \{2000, 4000, 6000, 8000, 10000\}$, the running times for DSCS are reported in Appendix E. Moreover, to further highlight the advantages of the DSCS algorithm, we selected three real-world Bayesian networks from the R-package bnlearn—WIN95PTS (76 nodes, 112 edges), LINK (724 nodes, 1125 edges), and MUNIN (1041 nodes, 1397 edges). A detailed description of the three Bayesian networks can be found at https://www.bnlearn.com/bnrepository/. For each network, we randomly sampled a vertex subset $M$ of size $|M|$, then applied DCR (on the DAGs) and DSCS (on the corresponding CPDAGs) to test collapsibility over $M$. The running times are also reported in Appendix E.

For $n \in \{500, 1000, 1500\}$, the average running times of the two algorithms are shown in Figure 3. For different values of $p$, the corresponding curves are distinguished by color and point shape. Detailed results, including average running times and standard errors, are also provided in Table E.1 in Appendix E. In Figure 3, across various graph sizes $n \in \{500, 1000, 1500\}$ and different values of $p$ and $|M|$, DSCS consistently outperforms DCR in terms of runtime. For a fixed $n$, the average running time of the DCR Algorithm increases rapidly with $p$ and $|M|$, while the DSCS Algorithm

exhibits much slower growth, demonstrating clear computational advantages. As $n$, $p$, and $|M|$ increase, the advantage of the DSCS Algorithm becomes more pronounced. For example, when $n = 500$, $p = 0.1$, and $|M| = 450$, the proposed DSCS is more than 530 times faster than DCR. When $n = 1500$, $p = 0.1$, and $|M| = 1350$, DSCS achieves a speedup of more than 4000 times.

## 5    Conclusion

In this work, we have tackled a fundamental bottleneck in high-dimensional Bayesian network inference: how to efficiently verify a submodel's estimation-collapsibility and reliably collapse away irrelevant variables. Departing from the traditional DAG framework [27, 28], we show that one can directly work on a CPDAG learned from observational data. We introduce the concept of sequential $c$-simplicity and find it as a necessary and sufficient criterion to check estimation-collapsibility. This criterion generalizes and unifies previous characterizations in both DAG models [28] and undirected chordal graph models [23, 25]. Remarkably, based on CPDAG, we are able to develop an order-free characterization (i.e. independent of the vertex permutation) and design an efficient Detecting Sequential $C$-simplicial Set (DSCS) algorithm. Beyond its theoretical value and computational efficiency, DSCS has profound practical consequences: it enables researchers to focus data-collection and computational effort on small subsets of interest to accelerate scientific discovery. We believe that incorporating collapsibility checks into researchers' preprocessing toolkit will significantly aid scalable probabilistic modeling, paving the way for robust and efficient inference in ever-expanding data regimes.

A limitation of this work is that we require the CPDAG to be accurately and fully learned from observational data. One could further relax the requirement of a fully accurate CPDAG by developing incremental or local CPDAG-learning strategies that suffice for collapsibility checks. Another direction is to extend DSCS to accommodate interventional data or dynamic Bayesian networks, broadening its applicability to sequential and adaptive decision-making.

## Acknowledgments and Disclosure of Funding

We thank the anonymous reviewers and area chair for their insightful comments, which helped strengthen this paper. This work was partially supported by the Key Program of the National Natural Science Foundation of China (Grant No.12431009) and the National Key Research and Development Program of China (Grant No.2020YFA0714100, Grant No.2020YFA0714102). The first two authors contribute equally. Shiyuan He was partially supported by the National Natural Science Foundation of China (Grant Numbers 12571278).

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

# A Graph terminology

A graph $\mathcal{G} = (V, E)$ is defined by a set $V$ of nodes (or vertices), and a set $E$ consisting of both directed and undirected edges. There is at most one edge between any pair of nodes $u$ and $v$. We use $(u, v)$ or $u \to v$ to denote a directed edge from $u$ to $v$, and $u - v$ to represent an undirected edge connecting $u$ and $v$. The nodes $u$ and $v$ are *adjacent* if there is an edge connecting them, which is denoted by $u \overset{\mathcal{G}}{\sim} v$. Otherwise, $u, v$ are nonadjacent and denoted by $u \overset{\mathcal{G}}{\nsim} v$. A *path* connecting $u$ and $v$ in $\mathcal{G}$, denoted by $\ell_{uv}$, is a sequence of distinct vertices $\langle c_0 = u, c_1, \ldots, c_{m-1}, c_m = v \rangle$ such that $c_i \overset{\mathcal{G}}{\sim} c_{i+1}$ for $i = 0, 1, \ldots, m - 1$. The path $\ell_{uv}$ is a *directed path*, denoted by $\vec{\ell}_{uv}$, from $u$ to $v$ if $c_i \to c_{i+1}$ for all $i = 0, 1, \ldots, m - 1$, and it is an *undirected path* if $c_i - c_{i+1}$ for all $i = 0, 1, \ldots, m - 1$. Additionally, the path $\ell_{uv}$ is a *partially directed path* from $u$ to $v$ if there are no edges of the form $c_i \leftarrow c_{i+1}$ for all $i = 0, 1, \ldots, m - 1$. A *cycle* is a path from a vertex to itself. An undirected path from $u$ to $v$, along with the edge $v - u$, forms an *undirected cycle*. A directed path from $u$ to $v$, along with the edge $v \to u$, forms a *directed cycle*. A *partially directed cycle* is formed by a partially directed path from $u$ to $v$, together with the edge $v \to u$ or $v - u$. We note that both directed paths (cycles) and undirected paths (cycles) are partially directed paths (cycles).

When $u \to v$, $u$ is referred to as the parent node of $v$, and $v$ is called the child node of $u$. If $u - v$, $u$ is a neighbor node of $v$. For some $u \in V$, let $\mathbf{pa}_{\mathcal{G}}(u)$ denote the set of parent nodes of $u$, $\mathbf{ch}_{\mathcal{G}}(u)$ the set of child nodes of $u$, and $\mathbf{ne}_{\mathcal{G}}(u)$ the set of neighbor nodes of $u$. Specifically, these sets are defined as follows:

$$\mathbf{pa}_{\mathcal{G}}(u) = \{w \in V \setminus \{u\} : w \to u \text{ in } \mathcal{G}\},$$
$$\mathbf{ch}_{\mathcal{G}}(u) = \{w \in V \setminus \{u\} : u \to w \text{ in } \mathcal{G}\},$$
$$\mathbf{ne}_{\mathcal{G}}(u) = \{w \in V \setminus \{u\} : u - w \text{ in } \mathcal{G}\}.$$

If $\mathbf{ch}_{\mathcal{G}}(u) = \emptyset$, then $u$ is called a *leaf node* in $\mathcal{G}$. For a subset $A \subseteq V$, the sets of its parents and children in $\mathcal{G}$ are, respectively, defined as

$$\mathbf{pa}_{\mathcal{G}}(A) \triangleq \bigcup_{u \in A} \mathbf{pa}_{\mathcal{G}}(u) \setminus A, \quad \mathbf{ch}_{\mathcal{G}}(A) \triangleq \bigcup_{u \in A} \mathbf{ch}_{\mathcal{G}}(u) \setminus A.$$

If $\mathbf{ch}_{\mathcal{G}}(A) = \emptyset$, then $A$ is called a *leaf set* in $\mathcal{G}$. We define the Markov boundary of $u$ and $A$ as $\mathbf{mb}_{\mathcal{G}}(u) = \mathbf{ch}_{\mathcal{G}}(u) \cup \mathbf{pa}_{\mathcal{G}}(u) \cup \mathbf{pa}_{\mathcal{G}}(\mathbf{ch}_{\mathcal{G}}(u)) \setminus \{u\}$ and $\mathbf{mb}_{\mathcal{G}}(A) = \bigcup_{u \in A} \mathbf{mb}_{\mathcal{G}}(u) \setminus A$.

For simplicity, we also define

$$\begin{aligned}
\mathbf{Pa}_{\mathcal{G}}(u) &= \mathbf{pa}_{\mathcal{G}}(u) \cup \{u\}, \quad \mathbf{Pa}_{\mathcal{G}}(A) = \mathbf{pa}_{\mathcal{G}}(A) \cup A; \\
\mathbf{Ch}_{\mathcal{G}}(u) &= \mathbf{ch}_{\mathcal{G}}(u) \cup \{u\}, \quad \mathbf{Ch}_{\mathcal{G}}(A) = \mathbf{ch}_{\mathcal{G}}(A) \cup A; \\
\mathbf{Ne}_{\mathcal{G}}(u) &= \mathbf{ne}_{\mathcal{G}}(u) \cup \{u\}, \quad \mathbf{Ne}_{\mathcal{G}}(A) = \mathbf{ne}_{\mathcal{G}}(A) \cup A; \\
\mathbf{Mb}_{\mathcal{G}}(u) &= \mathbf{mb}_{\mathcal{G}}(u) \cup \{u\}, \quad \mathbf{Mb}_{\mathcal{G}}(A) = \mathbf{mb}_{\mathcal{G}}(A) \cup A.
\end{aligned}$$

If there exists a directed path from $u$ to $v$ in $\mathcal{G}$, $u$ is referred to as an ancestor of $v$, and $v$ is called a descendant of $u$. The set of ancestors of $u$ in $\mathcal{G}$ is denoted as $\mathbf{an}_{\mathcal{G}}(u)$, and the set of descendants of $u$ in $\mathcal{G}$ is denoted as $\mathbf{de}_{\mathcal{G}}(u)$. Specifically, these sets are defined as follows:

$$\mathbf{an}_{\mathcal{G}}(u) = \left\{ w \in V \setminus \{u\} : \exists \vec{\ell}_{wu} \subseteq \mathcal{G} \right\}, \quad \mathbf{de}_{\mathcal{G}}(u) = \left\{ w \in V \setminus \{u\} : \exists \vec{\ell}_{uw} \subseteq \mathcal{G} \right\}.$$

For a subset $\mathbf{A} \subseteq V$, the sets of its ancestors and descendants in $\mathcal{G}$ are defined as follows:

$$\mathbf{an}_{\mathcal{G}}(A) \triangleq \bigcup_{u \in A} \mathbf{an}_{\mathcal{G}}(u) \setminus A, \quad \mathbf{de}_{\mathcal{G}}(A) \triangleq \bigcup_{u \in A} \mathbf{de}_{\mathcal{G}}(u) \setminus A.$$

We also define

$$\begin{aligned}
\mathbf{An}_{\mathcal{G}}(u) &= \mathbf{an}_{\mathcal{G}}(u) \cup \{u\}, \quad \mathbf{An}_{\mathcal{G}}(A) = \mathbf{an}_{\mathcal{G}}(A) \cup A; \\
\mathbf{De}_{\mathcal{G}}(u) &= \mathbf{de}_{\mathcal{G}}(u) \cup \{u\}, \quad \mathbf{De}_{\mathcal{G}}(A) = \mathbf{de}_{\mathcal{G}}(A) \cup A.
\end{aligned}$$

If $\mathcal{G}$ consists solely of undirected edges, it is referred to as an *undirected graph* (UG). In contrast, if $\mathcal{G}$ is composed entirely of directed edges and contains no directed cycles, it is referred to as a *directed acyclic graph* (DAG). Let $\vec{G} = (V, \vec{E})$ denote a DAG, where $V$ is the set of vertices and $\vec{E}$ is the set of directed edges in $\vec{G}$. A path $\ell_{uv}$ between $u$ and $v$ in $\vec{G}$ is said to be *blocked* by a set $S \subseteq V$ of vertices if and only if one of the following holds:

(*i*) $\ell_{uv}$ contains a *chain* $x \rightarrow z \rightarrow y$ or a *fork* $x \leftarrow z \rightarrow y$ such that the middle vertex $z$ is in $S$ (i.e., $z \in S$);

(*ii*) $\ell_{uv}$ contains a *collider* $x \rightarrow z \leftarrow y$ such that the middle vertex $z$ is not in $S$ and no descendant of $z$ is in $S$ (i.e., $\mathbf{De}_{\vec{G}}(z) \cap S = \emptyset$).

If all paths between $u$ and $v$ are blocked by $S$ in $\vec{G}$, then $u$ and $v$ are *d-separated* by $S$ in $\vec{G}$, denoted as $u \perp\!\!\!\perp v \mid S \, [\vec{G}]$. For disjoint subsets $A, B, S \subseteq \mathbf{V}$, if every path between any vertex $a \in A$ and any vertex $b \in B$ is blocked by $S$ in $\vec{G}$, then $A$ and $B$ are *d*-separated by $S$ in $\vec{G}$, denoted as $A \perp\!\!\!\perp B|S$ $[\vec{G}]$. The collection of all *d*-separation relations induced by $\vec{G}$ is denoted as

$$\mathcal{I}(\vec{G}) = \{\langle A, B|S\rangle : A \perp\!\!\!\perp B|S[\vec{G}] \text{ with pairwise disjoint subset } A, B, S \subseteq V\}.$$

For two directed acyclic graphs (DAGs) $\vec{G}_1 = (V, \vec{E}_1)$ and $\vec{G}_2 = (V, \vec{E}_2)$, if $\mathcal{I}(\vec{G}_1) = \mathcal{I}(\vec{G}_2)$, they are said to be *Markov equivalent*, denoted as $\vec{G}_1 \approx \vec{G}_2$. For three distinct vertices $u$, $v$, and $w$, if $u \rightarrow w \leftarrow v$ and $u$ is not adjacent to $v$ in $\vec{G}$, the triple $(u, w, v)$ is referred to as a *v-structure* collided on $w$. The *skeleton* of $\vec{G}$ is an undirected graph resulted from turning every directed edge in $\vec{G}$ into an undirected edge. Pearl et al. [55] showed that two DAGs are equivalent if and only if they share the same skeleton and the same *v*-structures. The *Markov equivalence class* of $\vec{G}$, or simply an *equivalence class*, denoted by $\mathcal{M}(\vec{G})$, contains all DAGs that are equivalent to $\vec{G}$. This equivalence class can be uniquely represented by a partially directed graph known as a *completely partially directed acyclic graph*(CPDAG) [45]. Let $G^* = (V, E^*)$ denote a CPDAG, where $V$ is the vertex set and $E^*$ is the set of directed and undirected edges of $G^*$. We use $\mathcal{M}(G^*)$ to denote the set of all Markov equivalent DAGs represented by the CPDAG $G^*$. It can be shown that the skeleton of a CPDAG $G^*$ is identical to the skeleton of every DAG in $\mathcal{M}(G^*)$, and an edge is directed in a CPDAG if and only if it is directed in every DAG in $\mathcal{M}(G^*)$ [55]. As demonstrated by Andersson et al.[46], $G^*$ is a chain graph composed of both directed and undirected edges, and it contains neither directed cycles nor partially directed cycles with directed edges. Additionally, the undirected components of a CPDAG $G^*$ are undirected and connected chordal graphs. A chordal graph (also known as a triangulated graph) is an undirected graph in which every cycle of length greater than three contains a chord—that is, an edge connecting two non-consecutive vertices in the cycle. Equivalently, a graph is chordal if and only if it contains no induced cycles of length greater than three.

## B    Estimation-collapsibility for UG models

For undirected graph models, estimation-collapsiblity has been characterized via simplicial set [50]. Asmussen & Edwards [23] and Frydenberg [25] established the equivalence between collapsibility over $M$ and the condition that each connected component of $M$ is a simplicial set, in the context of multinomial and Gaussian undirected graphical models. Below, we provide a brief introduction to the core concept of simplicial (vertex) sets in UG [50]. By drawing an analogy with the concept of sequentially *c*-removable sets in DAGs, we introduce the concept of sequential simplicial (vertex) sets in UG and explore its relationship with the concepts of simplicial (vertex) sets.

**Definition B.1.** *(Dirac [50]) In an undirected graph (UG) $G = (V, E)$, for a vertex $v \in V$, if the neighborhood $\mathbf{ne}_G(v)$ forms a complete subgraph in $G$, then $v$ is called a simplicial vertex in $G$. For a subset $M \subseteq V$, if the neighborhood $\mathbf{ne}_G(M)$ forms a complete subgraph in $G$, then $M$ is called a simplicial (vertex) set in $G$.*

Similar to the concept of *sequentially c-removable* in a DAG, we introduce the concept of *sequential simplicial (vertex) sets* in an undirected graph. An illustration of the concept is provided in Example B.1.

**Definition B.2.** *In an undirected graph $G = (V, E)$, for a subset $M \subseteq V$, if all vertices in $M$ can be ordered such that each vertex is a simplicial vertex according to that ordering, meaning each vertex can be identified as a simplicial vertex during successive removals, then $M$ is called a sequential simplicial (vertex) set in $G$.*

**Example B.1.** *Consider the UGs in Figures B.1 (a)–(c), and let's denote the three graphs as $G_1$, $G_2$, and $G_3$, respectively. For the $G_1$, we have $\mathbf{ne}_{G_1}(a) = \{b, c\}$ where $b \overset{G_1}{\sim} c$. By the Definition B.1,*

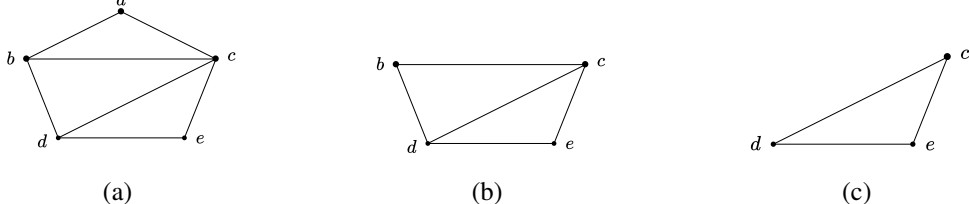

(a)            (b)            (c)

Figure B.1: *Simpliciality and sequential simpliciality.*

$a$ is a simplicial vertex in $G_1$. After removing $a$, the resulting UG $G_2$ has $\mathbf{ne}_{G_2}(b) = \{c, d\}$ with $c \overset{G_2}{\sim} d$. As a result, $b$ is a simplicial vertex in $G_2$. Similarly, after removing node $b$, we obtain the UG $G_3$, where $c$ is clearly a simplicial vertex in $G_3$. Therefore, the set $\{a, b, c\}$ is a sequential simplicial set in original UG $G_1$.

Note the classical concept of simplicial set and our proposed concept of sequential simplicial set are generally not equivalent. We illustrate this via Example B.2.

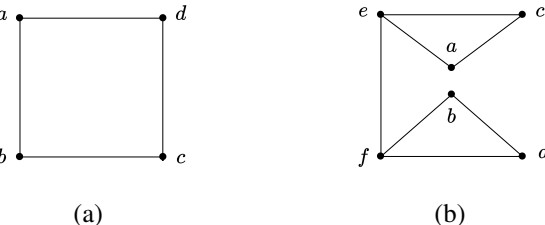

(a)            (b)

Figure B.2: *This example illustrates that the concepts of a simplicial set and a sequential simplicial set in undirected graphs are not equivalent. Figure B.2(a) shows that when $M = \{a, b\}$, $M$ is a simplicial set but not a sequential simplicial set. Figure B.2(b) shows that when $M = \{a, b\}$, $M$ is a sequential simplicial set but not a simplicial set.*

**Example B.2.** *In Figure B.2(a), let $M = \{a, b\}$, with $\mathbf{ne}_G(M) = \{c, d\}$ and $c \overset{G}{\sim} d$, which makes $M$ a simplicial set in $G$. However, neither $a$ nor $b$ is a simplicial vertex on its own, meaning that $M$ is not a sequential simplicial set. In Figure B.2(b), $M = \{a, b\}$, and both $a$ and $b$ are simplicial vertices in $G$, making $M$ a sequential simplicial set. However, $\mathbf{ne}_G(M) = \{c, d, e, f\}$, where $c \overset{G}{\not\sim} d$, $c \overset{G}{\not\sim} f$, and $d \overset{G}{\not\sim} e$, implying that $M$ is not a simplicial set in $G$.*

Under additional conditions, we can establish the equivalence between *simplicial sets* and *sequential simplicial sets* for undirected graphs. In Figure B.2(a), it is given that $M = \{a, b\}$ is a simplicial set in $G$, but $M$ is not a sequential simplicial set. However, if an additional condition such as $a \overset{G}{\sim} c$ or $b \overset{G}{\sim} d$ is imposed, then $M$ becomes a sequential simplicial set. In the more general case, the additional condition required is that $G_{\mathbf{Ne}_G(M)}$ is a chordal graph, as shown in Lemma B.1.

**Lemma B.1.** *Let $G = (V, E)$ is an undirected graph, for $M \subseteq V$, if $M$ is a simplicial set in $G$ and $G_{\mathbf{Ne}_G(M)}$ is a chordal graph, then $M$ is a sequential simplicial set in $G$.*

*Proof.* Here we consider using proof by contradiction to prove Lemma B.1. Let $M = \{m_1, m_2, \cdots, m_k\}$. Suppose that for any ordering, there exists a subsequence $m_i, m_{i+1}, \cdots, m_k$, where $1 \leq i \leq k$, such that none of these vertices are simplicial vertices in the induced subgraph $G_R \triangleq G_{V \setminus \{m_1, m_2, \cdots, m_{i-1}\}}$. This means that for all $m_j \in \{m_i, m_{i+1}, \cdots, m_k\}$, the neighborhood $\mathbf{ne}_{G_R}(m_j)$ is not complete, i.e., there exist vertices $x, y \in \mathbf{ne}_{G_R}(m_j)$ such that $x \overset{G}{\not\sim} y$. There are three possible cases:

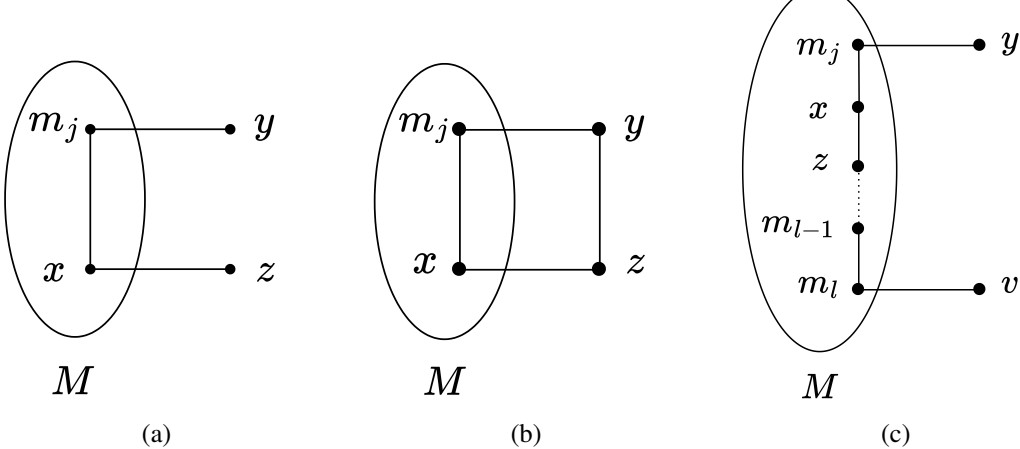

Figure B.3: *Three subgraphs of the UG. Figure B.3(a), Figure B.3(b), and Figure B.3(c) correspond to $(ia)$, $(ib)$, and $(ii)$ in case (2), respectively, in the proof of Lemma B.1.*

**Case (1)**. When $x, y \in V \setminus M$,

Since $x, y \in \mathbf{ne}_{G_R}(m_j)$ and $x, y \in V \setminus M$, it follows that $x, y \in \mathbf{ne}_G(M)$. However, $x \overset{G}{\not\sim} y$, which contradicts the fact that $M$ is a simplicial set in $G$.

**Case (2)**. When $x$ or $y$ is in $\{m_i, m_{i+1}, \cdots, m_k\} \setminus \{m_j\}$,
Without loss of generality, assume $x \in \{m_i, m_{i+1}, \cdots, m_k\} \setminus \{m_j\}$ and $y \in V \setminus M$. Since $x$ is not a simplicial vertex in $G_R$, there exists $z \in \mathbf{ne}_{G_R}(x)$ such that $z \overset{G}{\not\sim} m_j$.
$(i)$ If $z \in V \setminus M$, let's consider whether $y$ and $z$ are adjacent:

$(ia)$ If $y \overset{G}{\not\sim} z$, since $y, z \in ne_G(M)$, this contradicts the fact that $M$ is a simplicial set in $G$; see Figure B.3(a).
$(ib)$ If $y \overset{G}{\sim} z$, there exists a 4-cycle $m_j - x - z - y - m_j$, which contradicts the fact that $G_{\mathbf{Ne}_G(M)}$ is a chordal graph; see Figure B.3(b).

$(ii)$ If $z \in \{m_i, m_{i+1}, \cdots, m_k\} \setminus \{m_j, x\}$, since $z$ is also not a simplicial vertex in $G_R$, similar to the case with $x$, we need to analyze the adjacency of $z$'s neighbors. By applying this reasoning iteratively, we can construct a maximal connected component containing $m_j$ in the form $m_j - x - z - \cdots - m_{l-1} - m_l$. For this maximal connected component, there must exist $v \in \mathbf{ne}_{G_R}(m_l) \cap (V \setminus M)$ such that $v \overset{G}{\not\sim} m_{l-1}$ (otherwise, $m_l$ would be a simplicial vertex in $G_R$). This situation is analogous to case $(i)$, leading to a contradiction; see Figure B.3(c).

**Case (3)**. When $x, y \in (\{m_i, m_{i+1}, \cdots, m_k\} \setminus \{m_j\})$,
Since $x$ and $y$ are also not simplicial vertices in $G_R$, and given the arbitrary choice of $m_j$, $x$ and $y$ will encounter similar situations as in cases (1) and (2). $\quad\square$

Given the hereditary property of chordal graphs, any subgraph of a chordal graph is also a chordal graph. Therefore, for a chordal graph $G = (V, E)$, it is evident that $G_{\mathbf{Ne}_G(M)}$ is also chordal. Consequently, we have the following corollary.

**Corollary B.1.** *Given a chordal graph $G = (V, E)$, for $M \subseteq V$, if $M$ is a simplicial set in $G$, then $M$ is a sequential simplicial set in $G$.*

In Figure B.2(b), it is given that $M = \{a, b\}$ is a sequential simplicial set. However, $M$ is not a simplicial set in $G$. Nonetheless, if $M$ is a sequential simplicial set and is connected, then $M$ is a simplicial set in $G$.

**Lemma B.2.** *Given an undirected graph $G = (V, E)$, if $M \subseteq V$ is a connected subset and $M$ is a sequential simplicial set in $G$, then $M$ is a simplicial set in $G$.*

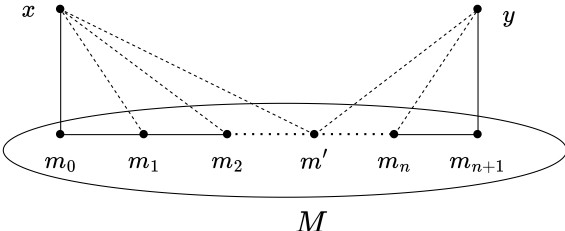

Figure B.4: *A subgraph of the UG. It corresponds to case(2) in the proof of Lemma B.2.*

*Proof.* Here we consider using proof by contradiction to prove Lemma B.2. Assume $M$ is not a simplicial set in $G$, i.e., there exist $x, y \in \mathbf{ne}_G(M)$ such that $x \overset{G}{\not\sim} y$. There are two possible cases:

**Case (1).** There exists $m \in M$ such that $x, y \in \mathbf{ne}_G(m)$.

Since $x \overset{G}{\not\sim} y$ and $x, y \in V \setminus M$, $m$ is not a simplicial vertex in $G$ for any ordering. This contradicts the assumption that $M$ is a sequential simplicial set in $G$.

**Case (2).** There exist $m_x, m_y \in M$ such that $x \in \mathbf{ne}_G(m_x)$ and $y \in \mathbf{ne}_G(m_y)$.

Since $m_x$ and $m_y$ are connected within $M$, there exists a chordless path $p = \langle m_0 = m_x, m_1, m_2, \cdots, m_n, m_{n+1} = m_y \rangle$ where $m_i \in M$ for $i \in [n]$. Because $p$ is chordless, for each $m_i$ (where $i \in [n]$), $m_{i-1}$ and $m_{i+1}$ are not adjacent, meaning $m_1, m_2, \cdots, m_n$ are not simplicial vertices. Since the vertices in $M$ can be ordered such that each vertex is a simplicial vertex according to that ordering, either $m_x$ or $m_y$ must be simplicial. Without loss of generality, assume $m_x$ is a simplicial vertex. Thus, $x \overset{G}{\sim} m_1$. Now consider the chordless path $p' = \langle m_1, m_2, \cdots, m_n, m_{n+1} = m_y \rangle$. In this case, $x \in \mathbf{ne}_G(m_1)$ and $y \in \mathbf{ne}_G(m_y)$. By similar analysis, iterating this process will lead to the existence of some $m' \in \{m_1, m_2, \cdots, m_n\}$ such that $x, y \in \mathbf{ne}_G(m')$, which reduces to case (1); see Figure B.4.

In both scenarios, we encounter contradictions. Therefore, the assumption that $M$ is not a simplicial set in $G$ must be false. Thus, if $M$ is a connected sequential simplicial set in $G$, then $M$ must be a simplicial set in $G$. $\qquad\qquad\square$

By Lemma B.2, we obtain a sufficient condition for $G_{\mathbf{Ne}_G(M)}$ to be chordal.

**Lemma B.3.** *Given an undirected graph $G = (V, E)$, if $M \subseteq V$ is a connected subset and $M$ is a sequential simplicial set in $G$, then $G_{\mathbf{Ne}_G(M)}$ is a chordal graph.*

*Proof.* By Lemma B.2, we know that $M$ is a simplicial set in $G$, which means $\mathbf{ne}_G(M)$ is a complete subgraph. Suppose $G_{\mathbf{Ne}_G(M)}$ is not a chordal graph. For any cycle $\mu$ in $G_{\mathbf{Ne}_G(M)}$ with length greater than 3, since $\mathbf{ne}_G(M)$ is complete, $\mu$ can contain at most two vertices from $\mathbf{ne}_G(M)$. Otherwise, $\mu$ would have a chord. Without loss of generality, assume that $\mu$ contains two vertices $x$ and $y$ from $\mathbf{ne}_G(M)$, and the remaining vertices are in $M$. Let $m_i$ be the first vertex removed from $\mu$, then $\mathbf{ne}_{G_\mu}(m_i)$ is complete. Thus, $\mu$ must have at least one chord, i.e., there exists an edge connecting two neighbors of $m_i$ in $\mu$. This contradicts the assumption that $\mu$ is a cycle of length greater than 3 without chords. Therefore, $G_{\mathbf{Ne}_G(M)}$ must be a chordal graph. $\qquad\square$

Based on the above series of lemmas, we can provide an equivalent characterization of a sequential simplicial set in an undirected graph that does not depend on the order, as stated in the following PropositionB.1.

**Proposition B.1.** *Given an undirected graph $G = (V, E)$, for $M \subseteq V$, the following two statements are equivalent:*

*(1) $M$ is a sequential simplicial set in $G$;*

*(2) For each connected component $\mathcal{C}_M$ of $M$, the subgraph $G_{\mathbf{Ne}_G(\mathcal{C}_M)}$ is chordal graph, and $\mathcal{C}_M$ is a simplicial set in $G$.*

*Proof. (1) $\Rightarrow$ (2).* Since $M$ is a sequential simplicial set in $G$, it is evident that $M_1, M_2, \cdots, M_K$ are also sequential simplicial sets in $G$. Therefore, for all $i \in [K]$, the vertices in $M_i$ can be ordered such that each vertex is a simplicial vertex according to that ordering. Additionally, since $M_i$ is connected, by Lemma B.2, $M_i$ is a simplicial set in $G$. By Lemma B.3, $G_{\mathbf{Ne}_G(M_i)}$ is a chordal graph.

*(2) $\Rightarrow$ (1).* Given that for all $i \in [K]$, $G_{\mathbf{Ne}(M_i)}$ is a chordal graph and $M_i$ is a simplicial set in $G$, by Lemma B.1, for all $i \in [K]$, the vertices in $M_i$ can be ordered such that they are simplicial vertices in that order, i.e., $M_i$ is a sequential simplicial set in $G$. Furthermore, since $M_1, M_2, \cdots, M_K$ are mutually disconnected, they can be ordered in any sequence, and they will still form a sequence of sequential simplicial sets. Therefore, $M$ is a sequential simplicial set in $G$. $\square$

For chordal graphs, by hereditary property, any subgraph of a chordal graph is also a chordal graph. Proposition B.1 immediately implies Lemma 1 for chordal graph. It states that in a chordal graph, for any connected subset, a sequential simplicial set is equivalent to a simplicial set. Therefore, for a chordal graph model, being estimation-collapsible over $M$ is equivalent to each connected subset of $M$ being a sequential simplicial set.

## C  Summary of Key Concepts

To facilitate comparison and reference, this appendix consolidates the key concepts related to simplicial, $c$-removable, and $c$-simplicial vertices and sets across UG, DAG, and CPDAG graphical models. The detailed definitions are presented in Table C.1.

Table C.1: Key Concepts in Graphical Models

| Graph Type | Concept | Characterization |
|---|---|---|
| UG | Simplicial vertex | A vertex $v$ is called *simplicial* if its neighborhood $\mathbf{ne}_G(v)$ forms a complete subgraph in $G$. |
| | Simplicial set | A set $M$, if $\mathbf{ne}_G(M)$ induces a complete subgraph, then $M$ is called a *simplicial set* in $G$. |
| DAG | $c$-removable vertex | A vertex $v$ is *c-removable* from DAG $\vec{G}$ if any two vertices in $\mathbf{Mb}_{\vec{G}}(v)$ are adjacent, except when both belong to $\mathbf{pa}_{\vec{G}}(v)$. |
| | Sequentially $c$-removable set | A set $M$ is *sequentially c-removable* if its vertices can be ordered such that each can be $c$-removed by that ordering. |
| CPDAG | $c$-simplicial vertex | A vertex $v$ is *c-simplicial* in $G^*$ if $\mathbf{ch}_{G^*}(v) = \emptyset$ and $\mathbf{ne}_{G^*}(v)$ is complete. |
| | Sequential $c$-simplicial set | A set $M$ is *sequential c-simplicial* in $G^*$ if its vertices can be successively removed as c-simplicial vertices in some order. |

The concept of a *c-simplicial vertex* in a CPDAG generalizes the notion of a *leaf node* (i.e., a node with an empty set of children) in DAGs and a *simplicial vertex* (i.e., a node whose set of neighbors is complete) [50] in undirected graphs. In particular, when $G^*$ is a DAG, $v$ is a *c-simplicial vertex* if and only if $v$ is a *leaf node*. This is because when $G^*$ is a DAG, it naturally holds that $\mathbf{ne}_{G^*}(v) = \emptyset$, making the completeness of its neighbor trivially satisfied, and the only condition required is $\mathbf{ch}_{G^*}(v) = \emptyset$. Meanwhile, when $G^*$ is an undirected graph, $v$ being a *c-simplicial vertex* in the CPDAG is equivalent to $v$ being a *simplicial vertex* in the undirected graph $G^*$. In this case, $\mathbf{ch}_{G^*}(v) = \emptyset$ always holds, and the sole requirement is that $\mathbf{ne}_{G^*}(v)$ forms a complete subgraph.

The notion of sequential $c$-simplicial set likewise generalizes sequential $c$-removable set for DAG. When $G^*$ is a DAG and $M$ is a sequential $c$-simplicial set, it directly follows from the previous

discussion that $M$ is also sequentially $c$-removable. Furthermore, we can find the concept of sequential $c$-simplicial set generalizes that of simplicial set for undirected graph. When $G^*$ contains only undirected edges (i.e. $G^*$ is a chordal graph), recall a $c$-simplicial vertex reduces to a simplicial vertex. This means that, for a sequential $c$-simplicial $M$, its vertices can be ordered such that each vertex is a simplicial vertex at the time of its removal. Consequently, in a chordal graph $G$, we also term a sequential $c$-simplicial set $M$ as a *sequential simplicial set*. Lemma 1 indicates that sequential simplicity (or sequential $c$-simplicity) is equivalent to simplicity for chordal graph.

## D   Technical Proofs of The Main Results

### Proof of Lemma 2

*Proof.* Assume that $\mathbf{ch}_{G^*}(M) \neq \emptyset$, then $\exists m \in M$ such that $\mathbf{ch}_{G^*}(m) \setminus M \neq \emptyset$. In this case, regardless of how the vertices in $M$ are ordered and sequentially removed as $c$-simplicial vertices, $m$ can never be a $c$-simplicial vertex. Therefore, by Definition 5, $M$ is not a sequential $c$-simplicial set in $G^*$, which contradicts the known result. $\square$

### Proof of Lemma 3

*Proof.* By Lemma 1, Statement $(2')$ is equivalent to:

$(2^*)$ $\mathbf{ch}_{G^*}(M) = \emptyset$, and $M$ is a sequential simplicial set in $G^*_{\tau_i}$.

We will now establish the equivalence between Statement (1) and Statement $(2^*)$.

*(1)$\Rightarrow$($2^*$).* Given that $\mathbf{ch}_{G^*}(M) = \emptyset$ clearly holds, we now need to prove that $M$ is a sequential simplicial set in $G^*_{\tau_i}$. Since $M$ is contained within a single chain component, and under the condition that $\mathbf{ch}_{G^*}(M) = \emptyset$, we have $\bigcup_{m \in M} \mathbf{ch}_{G^*}(m) = \emptyset$. Therefore, determining whether $M$ is a sequential simplicial set in $G^*_{\tau_i}$ is equivalent to determining whether $M$ is a sequential $c$-simplicial set in $G^*$.

*($2^*$)$\Rightarrow$(1).* Since $\mathbf{ch}_{G^*}(M) = \bigcup_{m \in M} \mathbf{ch}_{G^*}(m) \setminus M = \emptyset$, and because $M$ is contained within a single chain component, we have $\bigcup_{m \in M} \mathbf{ch}_{G^*}(m) = \emptyset$. In this case, $M$ *is a sequential simplicial set in* $G^*_{\tau_i}$ is clearly equivalent to $M$ *is a sequential $c$-simplicial set in* $G^*$. $\square$

### Proof of Theorem 2

*Proof.* By Lemma 1, Statement $(2')$ is equivalent to:

$(2^*)$ $\mathbf{ch}_{G^*}(M) = \emptyset$, where each $M_i$ forms a sequential simplicial set in $G^*_{\tau_i}$ for all $i \in [K]$.

We will now establish the equivalence between Statement (1) and Statement $(2^*)$.

*(1) $\Rightarrow$ ($2^*$).* $\mathbf{ch}_{G^*}(M) = \emptyset$ clearly holds. Next, we prove that for all $i \in [K]$, $M_i$ is a sequential simplicial set in $G^*_{\tau_i}$.
Without loss of generality, assume $\alpha(\tau_1) < \alpha(\tau_2) < \cdots < \alpha(\tau_K)$. Assume there exists a largest $j$, where $1 \leq j \leq K$, such that the non-empty $M_j$ is not a sequential simplicial set in $G^*_{\tau_j}$. Since $M$ is a sequential $c$-simplicial set in $G^*$, $M_K$ is also a sequential $c$-simplicial set in $G^*$. Let $G^*_1 \triangleq G^*_{V \setminus M_K}$. Then, it is evident that $M_{K-1}$ is a sequential $c$-simplicial set in $G^*_1$. Let $G^*_2 \triangleq G^*_{V \setminus \{M_K, M_{K-1}\}}$. Continuing this process, we deduce that $M_j$ is a sequential $c$-simplicial set in $G^*_{K-j} \triangleq G^*_{V \setminus \{M_K, M_{K-1}, \cdots, M_{j+1}\}}$. By Lemma 3, $M_j$ must be a sequential simplicial set in $G^*_{\tau_j}$, which contradicts our assumption.

*($2^*$) $\Rightarrow$ (1).* Without loss of generality, assume $\alpha(\tau_1) < \alpha(\tau_2) < \cdots < \alpha(\tau_K)$. Given that $\mathbf{ch}_{G^*}(M_K) = \emptyset$ and $M_K$ is a sequential simplicial set in $G^*_{\tau_K}$, by Lemma 3, $M_K$ is a sequential $c$-simplicial set in $G^*$. Let $G^*_1 \triangleq G^*_{V \setminus M_K}$. Then, $\mathbf{ch}_{G^*_1}(M_{K-1}) = \emptyset$ and $M_{K-1}$ is a sequential simplicial set in $G^*_{1, \tau_K}$. By Lemma 3 again, $M_{K-1}$ is a sequential $c$-simplicial set in $G^*_1$. Let $G^*_2 \triangleq G^*_{V \setminus \{M_K, M_{K-1}\}}$. Continuing this process, we find that $M_K, M_{K-1}, \cdots, M_1$ are sequential

$c$-simplicial sets. Therefore, the vertices in $M$ can be ordered such that they are $c$-simplicial vertices in $G^*$, meaning $M$ is a sequential $c$-simplicial set in $G^*$. $\qquad\square$

**Proof of Theorem 1**

*Proof. (1)$\Rightarrow$ (2).* Assume (2) does not hold. That is, for all $\vec{G} \in \mathcal{M}(G^*)$, $\mathbf{ch}_{\vec{G}}(M) \neq \emptyset$. Since $G^*$ is a CPDAG, this implies that $\mathbf{ch}_{G^*}(M) \neq \emptyset$ or there exists an undirected connected component $\mathcal{C}$ of $M$ such that $\mathbf{ne}_{G^*}(\mathcal{C})$ is not complete. By Theorem 2, $M$ is not a sequential $c$-simplicial set in $G^*$, which contradicts (1).

*(2)$\Rightarrow$ (1).* Let $\mathcal{T} = \{\tau_1, \tau_2, \ldots, \tau_K\}$ be the set of chain components of $G^*$. Partition $M$ into each chain component, i.e., $M = M_1 \cup M_2 \cup \cdots \cup M_K$, where $M_i \subseteq \tau_i$ for each $i \in [K]$. Assume (1) does not hold. By Theorem 2, there are two possible scenarios.

*(i)* $\mathbf{ch}_{G^*}(M) \neq \emptyset$. By the properties of CPDAGs, for all $\vec{G} \in \mathcal{M}(G^*)$, $\mathbf{ch}_{\vec{G}}(M) \neq \emptyset$, which contradicts (2).

*(ii)* There exists $j$, where $1 \leq j \leq K$, such that $M_j$ is not a sequential simplicial set in $G^*_{\tau_j}$. Let $M_j = M_{j,1} \cup M_{j,2} \cup \cdots \cup M_{j,l}$, where $M_{j,i}$ $(i \in [l])$ are the connected components in $G^*_{\tau_j}$. Since $M_j$ is not a sequential simplicial set in $G^*_{\tau_j}$ and $G^*_{\tau_j}$ is a chordal graph, by Lemma 1, there exists $i$, where $1 \leq i \leq l$, such that $M_{j,i}$ is not a simplicial set in $G^*$. That is, there exist $x, y \in \mathbf{ne}_{G^*_{\tau_j}}(M_{j,i})$ such that $x \overset{G^*}{\nsim} y$. In every DAG $\vec{G} \in \mathcal{M}(G^*)$, $x$ and $y$ cannot both be parents of $M_{j,i}$, otherwise a v-structure would be formed. Therefore, $\mathbf{ch}_{\vec{G}}(M_{j,i}) \neq \emptyset$, implying $\mathbf{ch}_{\vec{G}}(M) \neq \emptyset$, which contradicts (2).

*(2) $\Leftrightarrow$ (3)$\Leftrightarrow$ (4).* By [28, Lemma 1] and Theorem 1, we have (2) $\Leftrightarrow$ (3)$\Leftrightarrow$ (4) is evident. $\qquad\square$

# E   Additional Experimental Results

For the DSCS and DCR algorithms applied to random graphs with $n \in \{500, 1000, 1500\}$, the detailed average running times (in seconds) and the standard errors are presented in Table E.1. DSCS clearly shows a significant advantage over DCR, and this advantage becomes more pronounced as $n$ and $p$ increase. Furthermore, from Table E.1 and Figure 3, it can be observed that for a given $n$ and $|M|$, the average running time of DCR increases rapidly with increasing $p$, whereas DSCS exhibits a slower and approximately linear growth. Additionally, the standard errors of DSCS's average running times are consistently smaller than those of the DCR Algorithm, indicating greater stability in DSCS's running times across various random graphs.

Due to the extremely long computational time of the DCR Algorithm for large graphs, for $n \in \{2000, 4000, 6000, 8000, 10000\}$, we only record the average running time of the DSCS Algorithm. The results are shown in Table E.2 and in Figure E.1. It can be seen that even for high-dimensional variables (e.g., $n = 10000$) and denser graphs (e.g., $p = 0.1$), the DSCS Algorithm can quickly determine the estimation-collapsibility of the underlying DAG model. For example, when $n = 10000$, $p = 0.1$, and $|M| = 9000$, the average running time of the algorithm is just over 150 seconds. For a given $n$ and $p$, the average running time increases approximately linearly with $|M|$.

Based on these observations, it can be concluded that the DSCS Algorithm is capable of efficiently addressing the structure dimensionality reduction problem for high-dimensional Bayesian network with complex structure. It should be noted that some of the standard errors are relatively large due to the inherent randomness in the generation of the graphs and the random selection of the set $M$. For certain node subsets $M$, after finding $\mathbf{ch}_{G^*}(M) \neq \emptyset$ during the initial check (as per Lines 1–2 in the DSCS Algorithm), the DSCS algorithm can immediately stop, leading to a very fast determination of collapsibility. On the other hand, for some node subsets $M$ with $\mathbf{ch}_{G^*}(M) = \emptyset$, further checks are required (as per Lines 3–9 in the DSCS Algorithm), resulting in longer algorithm's running times.

Table E.1: The comparison of our method (DSCS) and detecting sequential $c$-removability (DCR) on $n = \{500, 1000, 1500\}$ in terms of average running time (in seconds) is shown below. The best result is bolded, and the values in parentheses represent the corresponding standard error of the mean.

| n = 500 | p = 0.1 | | p = 0.05 | | p = 0.01 | | p = 0.005 | |
|---|---|---|---|---|---|---|---|---|
| \|M\| | DSCS | DCR | DSCS | DCR | DSCS | DCR | DSCS | DCR |
| 50 | **0.037** (0.002) | 6.782 (0.368) | **0.036** (0.002) | 2.070 (0.108) | **0.038** (0.002) | 0.214 (0.017) | **0.045** (0.003) | 0.201 (0.017) |
| 100 | **0.084** (0.012) | 16.826 (0.591) | **0.067** (0.003) | 4.625 (0.219) | **0.086** (0.003) | 0.517 (0.023) | **0.078** (0.003) | 0.424 (0.015) |
| 150 | **0.131** (0.013) | 27.009 (0.918) | **0.096** (0.003) | 7.662 (0.331) | **0.114** (0.005) | 0.841 (0.028) | **0.127** (0.005) | 0.673 (0.019) |
| 200 | **0.164** (0.016) | 40.370 (1.724) | **0.143** (0.005) | 11.049 (0.491) | **0.158** (0.006) | 1.205 (0.031) | **0.170** (0.005) | 0.937 (0.023) |
| 250 | **0.209** (0.018) | 54.264 (2.449) | **0.171** (0.006) | 16.999 (0.608) | **0.207** (0.011) | 1.571 (0.045) | **0.220** (0.009) | 1.231 (0.033) |
| 300 | **0.260** (0.015) | 75.285 (3.780) | **0.200** (0.007) | 21.460 (0.834) | **0.246** (0.008) | 2.083 (0.042) | **0.263** (0.008) | 1.560 (0.043) |
| 350 | **0.285** (0.025) | 97.047 (4.818) | **0.215** (0.007) | 24.746 (0.787) | **0.277** (0.009) | 2.624 (0.050) | **0.315** (0.011) | 1.798 (0.041) |
| 400 | **0.320** (0.019) | 119.086 (5.475) | **0.264** (0.008) | 33.311 (1.150) | **0.333** (0.008) | 3.222 (0.063) | **0.347** (0.016) | 2.015 (0.058) |
| 450 | **0.318** (0.011) | 170.611 (10.245) | **0.336** (0.015) | 47.666 (1.794) | **0.376** (0.010) | 3.631 (0.098) | **0.414** (0.010) | 2.274 (0.057) |

| n = 1000 | p = 0.1 | | p = 0.05 | | p = 0.01 | | p = 0.005 | |
|---|---|---|---|---|---|---|---|---|
| \|M\| | DSCS | DCR | DSCS | DCR | DSCS | DCR | DSCS | DCR |
| 100 | **0.072** (0.003) | 73.394 (4.399) | **0.077** (0.005) | 33.503 (1.043) | **0.068** (0.003) | 0.795 (0.040) | **0.083** (0.011) | 0.444 (0.021) |
| 200 | **0.162** (0.022) | 155.271 (8.417) | **0.154** (0.008) | 71.075 (2.613) | **0.163** (0.006) | 1.862 (0.089) | **0.162** (0.009) | 1.026 (0.031) |
| 300 | **0.239** (0.015) | 288.592 (13.691) | **0.231** (0.011) | 124.544 (4.822) | **0.247** (0.010) | 3.099 (0.075) | **0.252** (0.009) | 1.803 (0.052) |
| 400 | **0.317** (0.029) | 407.435 (18.389) | **0.327** (0.020) | 211.905 (5.278) | **0.297** (0.010) | 4.915 (0.174) | **0.307** (0.009) | 2.499 (0.062) |
| 500 | **0.411** (0.036) | 536.017 (25.026) | **0.397** (0.011) | 264.091 (6.637) | **0.425** (0.017) | 6.534 (0.175) | **0.412** (0.014) | 3.533 (0.095) |
| 600 | **0.506** (0.042) | 680.526 (24.354) | **0.448** (0.013) | 311.422 (10.226) | **0.500** (0.015) | 8.765 (0.259) | **0.472** (0.011) | 4.504 (0.117) |
| 700 | **0.556** (0.034) | 850.131 (34.481) | **0.520** (0.010) | 398.958 (12.231) | **0.589** (0.010) | 10.998 (0.280) | **0.595** (0.015) | 5.639 (0.121) |
| 800 | **0.680** (0.040) | 1117.812 (46.555) | **0.618** (0.019) | 532.457 (13.992) | **0.633** (0.016) | 13.038 (0.384) | **0.675** (0.017) | 6.856 (0.200) |
| 900 | **0.957** (0.043) | 2261.976 (110.833) | **0.684** (0.022) | 690.801 (23.449) | **0.711** (0.018) | 16.399 (0.317) | **0.760** (0.019) | 8.240 (0.222) |

| n = 1500 | p = 0.1 | | p = 0.05 | | p = 0.01 | | p = 0.005 | |
|---|---|---|---|---|---|---|---|---|
| \|M\| | DSCS | DCR | DSCS | DCR | DSCS | DCR | DSCS | DCR |
| 150 | **0.133** (0.008) | 236.812 (13.180) | **0.118** (0.005) | 150.433 (6.096) | **0.157** (0.018) | 3.304 (0.122) | **0.125** (0.005) | 1.010 (0.042) |
| 300 | **0.324** (0.048) | 568.481 (29.277) | **0.249** (0.010) | 367.563 (11.262) | **0.244** (0.009) | 7.675 (0.212) | **0.250** (0.011) | 2.284 (0.072) |
| 450 | **0.440** (0.036) | 954.506 (48.465) | **0.350** (0.012) | 574.021 (25.087) | **0.416** (0.020) | 12.767 (0.295) | **0.399** (0.014) | 3.717 (0.074) |
| 600 | **0.520** (0.033) | 1332.058 (71.645) | **0.483** (0.013) | 812.180 (32.477) | **0.587** (0.030) | 17.383 (0.454) | **0.517** (0.011) | 5.968 (0.178) |
| 750 | **0.762** (0.025) | 2238.503 (152.182) | **0.619** (0.025) | 1133.943 (39.954) | **0.648** (0.015) | 24.295 (0.741) | **0.650** (0.014) | 8.123 (0.211) |
| 900 | **1.133** (0.064) | 3520.613 (180.584) | **0.748** (0.019) | 1523.037 (62.566) | **0.755** (0.022) | 32.174 (0.836) | **0.786** (0.019) | 10.265 (0.166) |
| 1050 | **1.215** (0.044) | 4341.776 (286.208) | **1.179** (0.024) | 2729.115 (106.079) | **0.897** (0.021) | 38.334 (1.004) | **0.907** (0.024) | 12.746 (0.241) |
| 1200 | **1.656** (0.096) | 5134.046 (283.055) | **1.286** (0.024) | 3417.224 (144.841) | **1.012** (0.023) | 49.865 (1.158) | **1.066** (0.029) | 15.470 (0.290) |
| 1350 | **2.015** (0.104) | 8190.708 (432.034) | **1.441** (0.050) | 4533.135 (251.822) | **1.143** (0.023) | 62.412 (1.634) | **1.211** (0.025) | 20.240 (0.439) |

Table E.2: The average running time (in seconds) of the DSCS Algorithm for $n = \{2000, 4000, 6000, 8000, 10000\}$ is shown below. The values in parentheses represent the corresponding standard error of the mean.

| n = 2000 | $|M|$ | | | | | | | | |
|---|---|---|---|---|---|---|---|---|---|
| $p$ | 300 | 500 | 700 | 900 | 1100 | 1300 | 1500 | 1700 | 1900 |
| 0.1 | 0.224 (0.007) | 0.467 (0.047) | 0.711 (0.074) | 0.941 (0.065) | 1.254 (0.079) | 1.476 (0.043) | 1.878 (0.111) | 2.286 (0.088) | 2.605 (0.086) |
| 0.05 | 0.212 (0.008) | 0.427 (0.045) | 0.604 (0.038) | 0.761 (0.038) | 0.985 (0.044) | 1.197 (0.047) | 1.486 (0.073) | 1.652 (0.051) | 1.884 (0.055) |
| 0.01 | 0.198 (0.008) | 0.333 (0.017) | 0.483 (0.008) | 0.627 (0.025) | 0.804 (0.013) | 0.907 (0.031) | 1.235 (0.034) | 1.342 (0.022) | 1.542 (0.066) |
| 0.005 | 0.193 (0.003) | 0.299 (0.012) | 0.488 (0.008) | 0.598 (0.028) | 0.884 (0.023) | 0.968 (0.015) | 1.144 (0.049) | 1.335 (0.033) | 1.526 (0.036) |

| n = 4000 | $|M|$ | | | | | | | | |
|---|---|---|---|---|---|---|---|---|---|
| $p$ | 600 | 1000 | 1400 | 1800 | 2200 | 2600 | 3000 | 3400 | 3800 |
| 0.1 | 2.067 (0.277) | 3.649 (0.276) | 9.452 (1.365) | 9.545 (0.423) | 11.170 (1.028) | 12.221 (0.468) | 13.542 (0.558) | 18.441 (0.611) | 18.644 (0.639) |
| 0.05 | 1.188 (0.069) | 2.445 (0.178) | 3.673 (0.198) | 5.992 (0.369) | 7.914 (0.369) | 8.525 (0.341) | 9.571 (0.300) | 10.378 (0.352) | 13.106 (0.419) |
| 0.01 | 0.785 (0.02) | 1.388 (0.037) | 1.940 (0.048) | 2.521 (0.047) | 3.167 (0.055) | 3.722 (0.055) | 4.515 (0.075) | 5.020 (0.070) | 5.185 (0.075) |
| 0.005 | 0.713 (0.013) | 1.118 (0.018) | 1.585 (0.019) | 2.110 (0.028) | 2.591 (0.038) | 3.172 (0.040) | 3.677 (0.046) | 3.983 (0.045) | 4.312 (0.033) |

| n = 6000 | $|M|$ | | | | | | | | |
|---|---|---|---|---|---|---|---|---|---|
| $p$ | 700 | 1300 | 1900 | 2500 | 3100 | 3700 | 4300 | 4900 | 5500 |
| 0.1 | 5.113 (0.719) | 13.039 (5.657) | 18.291 (1.918) | 16.860 (0.794) | 24.530 (1.671) | 28.101 (3.856) | 36.581 (1.203) | 33.083 (0.936) | 41.539 (0.768) |
| 0.05 | 3.833 (0.579) | 11.335 (0.769) | 6.426 (0.503) | 12.089 (1.115) | 11.453 (0.601) | 19.033 (1.839) | 21.954 (1.167) | 26.738 (1.708) | 33.046 (2.450) |
| 0.01 | 1.237 (0.051) | 2.665 (0.093) | 4.335 (0.235) | 5.629 (0.191) | 6.432 (0.275) | 10.354 (0.280) | 8.922 (0.254) | 10.107 (0.352) | 12.461 (0.491) |
| 0.005 | 1.093 (0.034) | 2.188 (0.060) | 3.133 (0.081) | 3.847 (0.063) | 4.351 (0.107) | 5.235 (0.096) | 5.993 (0.109) | 6.894 (0.106) | 7.912 (0.125) |

| n = 8000 | $|M|$ | | | | | | | | |
|---|---|---|---|---|---|---|---|---|---|
| $p$ | 1200 | 2000 | 2800 | 3600 | 4400 | 5200 | 6000 | 6800 | 7600 |
| 0.1 | 15.135 (1.681) | 35.980 (6.579) | 68.231 (2.654) | 51.077 (4.126) | 64.551 (14.213) | 62.069 (8.846) | 89.584 (27.189) | 145.404 (16.038) | 179.517 (13.521) |
| 0.05 | 10.916 (1.363) | 14.425 (1.165) | 38.392 (10.641) | 34.438 (1.347) | 36.200 (1.150) | 62.324 (10.026) | 67.810 (5.679) | 84.859 (10.766) | 87.497 (3.082) |
| 0.01 | 3.725 (0.276) | 6.715 (0.318) | 8.771 (0.501) | 12.433 (0.990) | 15.978 (0.893) | 20.836 (0.968) | 29.116 (1.401) | 30.605 (2.686) | 38.041 (5.092) |
| 0.005 | 3.077 (0.255) | 5.765 (0.272) | 4.979 (0.091) | 6.670 (0.097) | 9.418 (0.747) | 11.517 (0.908) | 15.398 (2.049) | 13.895 (0.370) | 15.198 (0.179) |

| n = 10000 | $|M|$ | | | | | | | | |
|---|---|---|---|---|---|---|---|---|---|
| $p$ | 1000 | 2000 | 3000 | 4000 | 5000 | 6000 | 7000 | 8000 | 9000 |
| 0.1 | 8.344 (1.881) | 48.183 (24.965) | 47.575 (12.945) | 48.310 (6.958) | 79.901 (21.997) | 168.653 (60.873) | 177.600 (58.726) | 143.666 (18.392) | 157.696 (14.677) |
| 0.05 | 19.787 (2.858) | 31.385 (8.300) | 44.263 (3.244) | 61.835 (2.463) | 75.421 (3.410) | 99.957 (8.690) | 158.791 (23.000) | 169.996 (18.396) | 100.834 (2.665) |
| 0.01 | 4.197 (0.346) | 7.108 (0.263) | 9.011 (0.411) | 12.373 (1.468) | 14.324 (0.948) | 21.176 (2.523) | 22.330 (1.223) | 29.326 (2.701) | 30.720 (1.408) |
| 0.005 | 3.229 (0.253) | 3.569 (0.065) | 7.257 (0.141) | 11.486 (0.161) | 12.277 (0.309) | 14.850 (0.237) | 19.788 (0.349) | 24.472 (0.270) | 30.169 (1.331) |

To further highlight the advantages of the DSCS algorithm, we selected three real-world Bayesian networks from the R-package bnlearn—WIN95PTS (76 nodes, 112 edges), LINK (724 nodes, 1125 edges), and MUNIN (1041 nodes, 1397 edges). A detailed description of the three Bayesian networks can be found at https://www.bnlearn.com/bnrepository/. For each network, we randomly sampled a vertex subset $M$ of size $|M|$, then applied DCR (on the DAGs) or DSCS (on the corresponding CPDAGs) to test estimate-collapsibility over $M$. For every combination of network

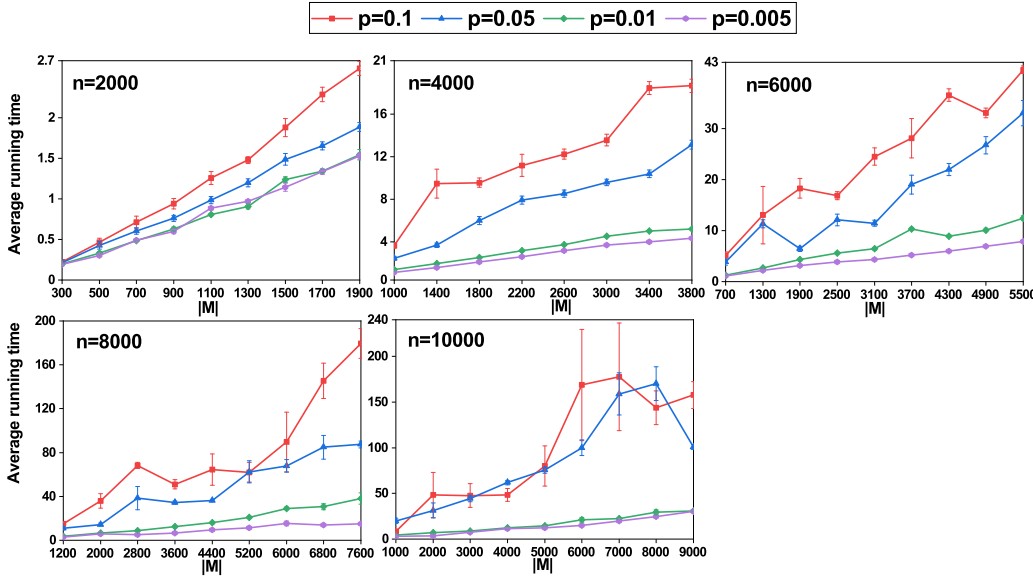

Figure E.1: The average running time of the DSCS Algorithm for different $n$ and $p$ as a function of $|M|$.

and $|M|$, we ran 30 simulations and recorded the average running time (in seconds), as reported in the Table E.3.

Table E.3: Comparison of average running time (seconds) between DSCS and DCR methods on three real-world Bayesian networks (the best result is bolded).

| WIN95PTS: $|V| = 76, |\vec{E}| = 112$ | | | | | | | | |
|---|---|---|---|---|---|---|---|---|
| $|M|$ | 5 | 13 | 21 | 29 | 37 | 45 | 53 | 61 | 69 |
| DSCS | **0.003** | **0.007** | **0.012** | **0.019** | **0.023** | **0.026** | **0.031** | **0.035** | **0.047** |
| DCR | 0.020 | 0.040 | 0.067 | 0.104 | 0.144 | 0.189 | 0.255 | 0.289 | 0.333 |

| LINK: $|V| = 724, |\vec{E}| = 1125$ | | | | | | | | |
|---|---|---|---|---|---|---|---|---|
| $|M|$ | 50 | 130 | 210 | 290 | 370 | 450 | 530 | 610 | 690 |
| DSCS | **0.027** | **0.072** | **0.114** | **0.168** | **0.238** | **0.343** | **0.442** | **0.596** | **0.660** |
| DCR | 0.141 | 0.449 | 0.830 | 1.359 | 2.199 | 3.253 | 4.425 | 6.147 | 7.024 |

| MUNIN: $|V| = 1041, |\vec{E}| = 1397$ | | | | | | | | |
|---|---|---|---|---|---|---|---|---|
| $|M|$ | 100 | 200 | 300 | 400 | 500 | 600 | 700 | 800 | 900 |
| DSCS | **0.058** | **0.108** | **0.182** | **0.260** | **0.383** | **0.511** | **0.710** | **0.742** | **0.983** |
| DCR | 0.333 | 0.771 | 1.588 | 2.822 | 4.168 | 6.762 | 9.260 | 11.954 | 15.420 |

Table E.3 demonstrates the decisive superiority of the proposed DSCS method over the DCR baseline in computational efficiency. Across all three real-world Bayesian networks, DSCS consistently achieves significantly lower running times for every tested size of the query set $|M|$. The performance gap is particularly evident on the largest network (MUNIN), where at $|M| = 900$, DSCS completed in under a second while DCR required over 15 seconds. These results robustly confirm that DSCS is a more efficient and scalable solution for determining estimation-collapsibility, making it highly suitable for application on high-dimensional Bayesian networks.

