# OpenReview forum: "DSCS: Fast CPDAG-Based Verification of Collapsible Submodels in High-Dimensional Bayesian Networks"
_NeurIPS.cc/2025/Conference — NeurIPS 2025 poster_

### Official Review · Reviewer_QbiJ · 2025-06-11

**Clarity:** 3
**Significance:** 3
**Originality:** 3
**Rating:** 5
**Confidence:** 3

**Summary:**

The paper studies the problem of when inference and marginalization in Bayesian networks can be performed by simply looking at the subgraph induced by the variables of interest rather than having to consider all variables. The authors design a generalization of existing algorithms for DAGs and undirected graphs by considering the completed partially directed acyclic graph (CPDAG) that encodes the Markov equivalence class of the Bayesian network. The main idea is to take advantage of the fact that CPDAGs are chains of undirected connected chordal graphs (UCCGs), since the authors observe that testing the property within UCCGs is straightforward. The running time is only quadratic in the number of vertices whereas previous work on DAGs required $O(|V|^4)$ time, and the speedup is evident is also from the empirical experiments.

**Questions:**

You write that "Subsequent parallel verification of neighborhood completeness across chain components has complexity of order $O(|\tau_{\max}|^2)$, dominated by the largest component". If I understood correctly, the issue is that you take quadratic time to evaluate whether each neighborhood is a clique? Have you considered if this could be done in linear time in the number of edges? I suspect that this might be doable by computing a perfect elimination ordering for each chordal component in linear time, since these enable characterizing the maximal cliques of the graph (see, e.g., [1]). Then, you would probably just need to check whether the neighborhood is a subset of the maximal clique associated with the node of the neighborhood that appears in the elimination ordering the earliest.

[1] Fanica Gavril:
Algorithms for Minimum Coloring, Maximum Clique, Minimum Covering by Cliques, and Maximum Independent Set of a Chordal Graph. SIAM J. Comput. 1(2): 180-187 (1972)

**Ethical Concerns:**

["NO or VERY MINOR ethics concerns only"]

**Final Justification:**

The Authors have adequately addressed the other Reviewers' and my concerns, so I continue to think that this is solid work to be accepted. I consider the additional experiments that were performed for the rebuttal especially beneficial for the work, since the earlier experiments relying only on Erdös-Rényi were too limited in my opinion.

**Limitations:**

yes

**Quality:**

3

**Strengths And Weaknesses:**

**Strengths:**

Quadratic speedup in testing estimation-collapsibility for DAGs.

Experiments providing practical evidence on the speedup.

Concrete examples that are associated with illustrations to assist the reader.

**Weaknesses:**

I think it would be desirable for the main manuscript to be self-contained; now, some terminology is defined only in the Appendix (see, e.g., Sec. 2.1, and related minor comments). Some further polishing could also be done (see minor comments for a non-comprehensive list of examples).

The experiments only consider Erdös-Rényi graphs and not any real-world instances.

**Minor:**
- Line 49: missing whitespace.
- Line 92: $A$ is already a set, so it does not need braces around it.
- Line 99: missing whitespace.
- Fig. 2 caption: missing whitespace.
- Line 121: grammatical error in "a substantially saving of data collection and computational efforts".
- Lines 152, 153: maybe "induces" instead of "forms"?
- Definition 5: I think you should emphasize that the endpoints of undirected edges of a CPDAG do not count as children. Similarly, it is unclear that children or parents aren't neighbors unless one reads the Appendix containing the definitions carefully.
- Line 191: you have a set instead of an ordering.
- Line 192: I think it is more common to call childless nodes sinks rather than leafs.
- Line 265: Extra 'p'.
- Line 529: The definition of a chord could be more precise ("edge connecting two non-adjacent nodes") since the nodes are, by definition, adjacent if the edge connects them. Alternatives include "no induced cycle of length greater than 3" or "edge connecting two non-consequent nodes of the cycle".

---

> ### Author Rebuttal · Authors · 2025-07-29
>
> We sincerely thank the reviewer for your careful reading and valuable feedback, which will help to significantly improve the work.
>
> **Weaknesses**
> > I think it would be desirable for the main manuscript to be self-contained; now, some terminology is defined only in the Appendix (see, e.g., Sec. 2.1, and related minor comments). Some further polishing could also be done (see minor comments for a non-comprehensive list of examples).
> The experiments only consider Erdös-Rényi graphs and not any real-world instances.
>
> Thank you for your careful reading and for correcting the formatting, grammar, and spelling. We will thoroughly proofread the entire manuscript again.
>
> Most concepts have been put in the Appendix due to space limitations. To make the main manuscript self-contained, we will add more detailed explanations for some important terminologies in the final version of the manuscript.
>
> Meanwhile, we have now selected three real-world Bayesian networks from the R package *bnlearn*—WIN95PTS (76 nodes, 112 edges), LINK (724 nodes, 1125 edges), and MUNIN (1041 nodes, 1397 edges). For each network, we randomly sampled a vertex subset $M$ of size $|M|$, then applied DCR (on the DAGs) or DSCS (on the corresponding CPDAGs) to test estimate-collapsibility over $M$. For every combination of network and $|M|$, we ran 30 simulations and recorded the average running time (in seconds), as reported in the table below. Across all graphs and values of $|M|$, DSCS consistently outperforms DCR in runtime, with its advantage growing as both the network size and $|M|$ increase.
>
> **Table 1:** *Comparison of average running time (seconds) between DSCS and DCR methods (the best result is bolded).*
>
> **WIN95PTS**: \|$V$\| = 76, \|$\vec{E}$\| = 112
> | | | | | | | | | | |
> |:----------:|:---:|:---:|:---:|:---:|:---:|:---:|:---:|:---:|:---:|
> | \|$M$\| |   5   | 13    | 21    | 29    | 37    | 45    | 53    | 61    | 69    |
> | DSCS  | **0.003** |**0.007** | **0.012** | **0.019** | **0.023** | **0.026** | **0.031** | **0.035** | **0.047** |
> | DCR   | 0.020 | 0.040 | 0.067 | 0.104 | 0.144 | 0.189 | 0.255 | 0.289 | 0.333 |
> | | | | | | | | | | |
>
> **LINK**: \|$V$\| = 724, \|$\vec{E}$\| = 1125
> | | | | | | | | | | |
> |:----------:|:---:|:---:|:---:|:---:|:---:|:---:|:---:|:---:|:---:|
> | \|$M$\| | 50 | 130 | 210 | 290 | 370 | 450 | 530 | 610 | 690 |
> | DSCS | **0.027** | **0.072** | **0.114** | **0.168** | **0.238** | **0.343** | **0.442** | **0.596** | **0.660** |
> | DCR | 0.141 | 0.449 | 0.830 | 1.359 | 2.199 | 3.253 | 4.425 | 6.147 | 7.024 |
> | | | | | | | | | | |
>
> **MUNIN**: \|$V$\| = 1041, \|$\vec{E}$\| = 1397
> | | | | | | | | | | |
> |:----------:|:---:|:---:|:---:|:---:|:---:|:---:|:---:|:---:|:---:|
> |\|$M$\| | 100 | 200 | 300 | 400 | 500 | 600 | 700 | 800 | 900 |
> | DSCS | **0.058** | **0.108** | **0.182** | **0.260** | **0.383** | **0.511** | **0.710** | **0.742** | **0.983** |
> | DCR | 0.333 | 0.771 | 1.588 | 2.822 | 4.168 | 6.762 | 9.260 | 11.954 | 15.420 |
> | | | | | | | | | | |
>
>
> These additional experiments will be included in the Appendix to further support the proposed method.
>
> &nbsp;
>
> **Questions**
> > You write that "Subsequent parallel verification of neighborhood completeness across chain components has complexity of order $O(|\tau_{\max}|^2)$, dominated by the largest component". If I understood correctly, the issue is that you take quadratic time to evaluate whether each neighborhood is a clique? Have you considered if this could be done in linear time in the number of edges? I suspect that this might be doable by computing a perfect elimination ordering for each chordal component in linear time, since these enable characterizing the maximal cliques of the graph (see, e.g., [1]). Then, you would probably just need to check whether the neighborhood is a subset of the maximal clique associated with the node of the neighborhood that appears in the elimination ordering the earliest.
> [1] Fanica Gavril: Algorithms for Minimum Coloring, Maximum Clique, Minimum Covering by Cliques, and Maximum Independent Set of a Chordal Graph. SIAM J. Comput. 1(2): 180-187 (1972)
>
> Thank you very much for your constructive suggestion, and your understanding is absolutely correct. We carefully reviewed the literature (see, e.g., Fanica 1972) you provided as well as subsequent works (see, e.g., Rose et al. 1976; Tarjan & Yannakakis 1984), and indeed, there exists an algorithm called the *maximum cardinality search* algorithm (MCS) (see, e.g., Rose et al. 1976; Tarjan & Yannakakis 1984), which can find a perfect elimination ordering of chordal graphs while identifying all maximal cliques in the graph, with linear complexity. Therefore, we can use the MCS algorithm to determine whether each neighborhood is a clique, as you mentioned, by checking whether the neighborhood is a subset of the maximal clique associated with the node of the neighborhood that appears in the elimination ordering the earliest. Consequently, the complexity of Line 8 in the DSCS algorithm is no longer $O(|\tau\_{max}|^2)$, but $O(|\tau_{max}|+|E^{\*}\_{\tau_{max}}|)$. Thus, the overall complexity of the DSCS algorithm is $O(|V|+|E^{\*}|)$, and we will make sure to highlight this in the main text. Once again, thank you for providing such valuable and insightful suggestions.
>
> **Reference**
>
> Fanica Gavril: Algorithms for Minimum Coloring, Maximum Clique, Minimum Covering by Cliques, and Maximum Independent Set of a Chordal Graph. SIAM J. Comput. 1(2): 180-187 (1972)
>
> Rose D J, Tarjan R E, Lueker G S. Algorithmic aspects of vertex elimination on graphs[J]. SIAM Journal on computing, 1976, 5(2): 266-283.
>
> Tarjan R E, Yannakakis M. Simple linear-time algorithms to test chordality of graphs, test acyclicity of hypergraphs, and selectively reduce acyclic hypergraphs[J]. SIAM Journal on computing, 1984, 13(3): 566-579.

---

> > ### Comment · Reviewer_QbiJ · 2025-08-01
> >
> > Thank you for the rebuttal and the additional experiments! I'm satisfied by the response and maintain my score.

---

### Official Review · Reviewer_DeqM · 2025-07-03

**Clarity:** 4
**Significance:** 4
**Originality:** 4
**Rating:** 5
**Confidence:** 4

**Summary:**

This work aims to propose efficient algorithms to verify estimation-collapsibility on a completed partially directed acyclic graph, or CPDAG, for Bayesian network inference. It introduces the concept of sequential c-simplicial sets and reveals the connection between the sequential c-simplicial sets and estimation-collapsibility. Further, it proposes a so-called detecting sequential c-simplicial sets (DSCS) algorithm as the verification algorithm with computational complexity analysis to show its efficiency. Empirical results on synthetic data are further presented.

**Questions:**

- What is the complexity of the conversion from a DAG of a Bayesian network to a CPDAG?
- Can the authors provide more quantitative or qualitative comparisons with other baseline methods?

**Ethical Concerns:**

["NO or VERY MINOR ethics concerns only"]

**Final Justification:**

The authors have addressed my concerns. So I keep my positive score.

**Limitations:**

Yes.

**Paper Formatting Concerns:**

None.

**Quality:**

4

**Strengths And Weaknesses:**

[strength]
- The research question considered in this work is important and challenging since the estimation-collapsibility can greatly improve the inference efficiency for many probabilistic queries.
- The introduced concept of sequential c-simplicial sets and its connection with the estimation-collapsibility is novel, which unifies the previous work.
- This work is overall well written, with a thorough discussion on previous work and theoretical results clearly elaborated. While this work is heavy in notations, I find the running example using Figure 2 very helpful.

[weakness]
- The experimental section will benefit from including more baseline methods for comparison, since now only one baseline is considered. It will also be improved a lot if more benchmarks are included, especially the real-world ones and those with diverse graph structures, since currently only results on Bayesian networks with Erdos-Renyi graphs are presented.

---

> ### Author Rebuttal · Authors · 2025-07-29
>
> Thank you for your careful reading and valuable feedback. To make the experimental section more comprehensive and persuasive, we will add experimental results on real-world graph structures from the bnlearn—Bayesian Network Repository in the Appendix. Regarding the questions you raised, we address them in detail below.
>
>
>
> **Question 1**
> > What is the complexity of the conversion from a DAG of a Bayesian network to a CPDAG?
>
> Thank you for your raising this important question. For a given DAG, the complexity of converting it to a CPDAG is $O(|V|^3)$ (see e.g., Chickering 2002). Therefore, even if the underlying DAG is known, it can still be converted to a CPDAG, and then the proposed DSCS algorithm can be applied with $O(|V|^2)$. Thus, the overall procedure has complexity $O(|V|^3)$. Our method improves upon the naive $O(|V|^4)$ approach and, to our best knowledge, represents the most efficient algorithm available for this problem. We will clarify this point  in the revised manuscript.
>
> **Reference**
>
>  Chickering D M. Optimal structure identification with greedy search[J]. Journal of Machine Learning Research, 2002, 3(Nov): 507-554.
>
>  &nbsp;
>
> **Question 2** and **Weaknesses**
> > The experimental section will benefit from including more baseline methods for comparison, since now only one baseline is considered. It will also be improved a lot if more benchmarks are included, especially the real-world ones and those with diverse graph structures, since currently only results on Bayesian networks with Erdos-Renyi graphs are presented.
>
> As far as we know, the methods in [28] we compare against are the optimal approaches for addressing DAGs in the literature, and the current work is the first to investigate this problem based on CPDAGs. Therefore, we are unable to compare with additional methods.
>
> Meanwhile, we have now selected three real-world Bayesian networks from the R package *bnlearn*—WIN95PTS (76 nodes, 112 edges), LINK (724 nodes, 1125 edges), and MUNIN (1041 nodes, 1397 edges). For each network, we randomly sampled a vertex subset $M$ of size $|M|$, then applied DCR (on the DAGs) or DSCS (on the corresponding CPDAGs) to test estimate-collapsibility over $M$. For every combination of network and $|M|$, we ran 30 simulations and recorded the average running time (in seconds), as reported in the table below. Across all graphs and values of $|M|$, DSCS consistently outperforms DCR in runtime, with its advantage growing as both the network size and $|M|$ increase.
>
> **Table 1:**  *Comparison of average running time (seconds) between DSCS and DCR methods (the best result is bolded).*
>
> **WIN95PTS**: \|$V$\| = 76, \|$\vec{E}$\| = 112
> | | | | | | | | | | |
> |:----------:|:---:|:---:|:---:|:---:|:---:|:---:|:---:|:---:|:---:|
> | \|$M$\| |   5   | 13    | 21    | 29    | 37    | 45    | 53    | 61    | 69    |
> | DSCS  | **0.003** |**0.007** | **0.012** | **0.019** | **0.023** | **0.026** | **0.031** | **0.035** | **0.047** |
> | DCR   | 0.020 | 0.040 | 0.067 | 0.104 | 0.144 | 0.189 | 0.255 | 0.289 | 0.333 |
> | | | | | | | | | | |
>
> **LINK**: \|$V$\| = 724, \|$\vec{E}$\| = 1125
> | | | | | | | | | | |
> |:----------:|:---:|:---:|:---:|:---:|:---:|:---:|:---:|:---:|:---:|
> | \|$M$\| | 50 | 130 | 210 | 290 | 370 | 450 | 530 | 610 | 690 |
> | DSCS | **0.027** | **0.072** | **0.114** | **0.168** | **0.238** | **0.343** | **0.442** | **0.596** | **0.660** |
> | DCR | 0.141 | 0.449 | 0.830 | 1.359 | 2.199 | 3.253 | 4.425 | 6.147 | 7.024 |
> | | | | | | | | | | |
>
> **MUNIN**: \|$V$\| = 1041, \|$\vec{E}$\| = 1397
> | | | | | | | | | | |
> |:----------:|:---:|:---:|:---:|:---:|:---:|:---:|:---:|:---:|:---:|
> |\|$M$\| | 100 | 200 | 300 | 400 | 500 | 600 | 700 | 800 | 900 |
> | DSCS | **0.058** | **0.108** | **0.182** | **0.260** | **0.383** | **0.511** | **0.710** | **0.742** | **0.983** |
> | DCR | 0.333 | 0.771 | 1.588 | 2.822 | 4.168 | 6.762 | 9.260 | 11.954 | 15.420 |
> | | | | | | | | | | |
>
> These additional experiments will be included in the Appendix to further support the proposed method.

---

> > ### Comment · Reviewer_DeqM · 2025-08-05
> >
> > Thanks for your reply. I will keep my positive score.

---

### Official Review · Reviewer_ufyg · 2025-07-04

**Clarity:** 3
**Significance:** 3
**Originality:** 3
**Rating:** 4
**Confidence:** 4

**Summary:**

This paper studies the problem of estimation collapsibility that is fitting a sub-model and ignoring non-essential variables. This is relevant in scenarios where the number of variables are much more than the relevant ones. The authors introduce sequential c-simplicial sets to characterize estimation collapsibility in CPDAG instead of DAG and introduce an algorithm for verifying estimation-collapsibility within CPDAGs.

**Questions:**

Please see the comments.

**Ethical Concerns:**

["NO or VERY MINOR ethics concerns only"]

**Final Justification:**

The authors rebuttal addresses some of my concerns and thus my recommendation is toward accept.

**Limitations:**

Please see the comments.

**Paper Formatting Concerns:**

No major issues.

**Quality:**

3

**Strengths And Weaknesses:**

Strength:
As mentioned, often in practice, the exact DAG is not available and using observational data, only up to Marko equivalence class can be revealed. Hence, the result of this work which characterises the estimation collapsibility for CPDAGs becomes important.

The proposed algorithm has complexity of quadratic from in the number of vertices which makes it scalable for large networks.

Weakness:
The paper’s presentation has room for improvement. Some important definitions are missing such as non-triviality or chain components.
Is the chain components are unique? If not how large the set of components, i.e., what is K in line 3 of DSCS algorithm?
Since chain components are a major part of the proposed approach without (I could not find it) proper definitions and discussion on its cardinality, it is hard to read and understand the difference between the proposed method and the previous ones.
Why for example, other methods do not use the chain components Idea and reduce their complexity?


Although the result of this work for CPDAGs are novel but given the previous definitions and characterisation of the c-removability, it is a limited contribution. In other words, it seems that given the previous results of estimation collapsibility in DAGs, its extension to CPDAGs is not trivial but a natural generalisation.

---

> ### Author Rebuttal · Authors · 2025-07-29
>
> We sincerely thank the reviewer for your careful reading and thoughtful feedback. Your comments will help us further clarify and highlight the novel contributions of our work. To improve the readability of the paper, we will also elaborate on key definitions in the final manuscript. In response to your comments, we address each point in detail below.
>
> **Weakness 1**
> >The paper's presentation has room for improvement. Some important definitions are missing such as non-triviality or chain components. Is the chain components are unique? If not how large the set of components, i.e., what is K in line 3 of DSCS algorithm? Since chain components are a major part of the proposed approach without (I could not find it) proper definitions and discussion on its cardinality, it is hard to read and understand the difference between the proposed method and the previous ones. Why for example, other methods do not use the chain components Idea and reduce their complexity?
>
> Thank you for the valuable feedback. We agree that precise definitions and a clear discussion of chain components and non-triviality are crucial for understanding our approach. We will ensure that all such concepts are explicitly defined in the revised manuscript.
>
> Regarding ***non-triviality***, as stated in Line 131, this assumption was originally proposed in [28] and imposes only mild distributional constraints. It is pointed out in  [28] and we quote here that: "the DAG models for both contingency tables and the Gaussian distributions are non-trivial," and furthermore, "non-triviality is satisfied by most DAG models and is also implicitly used in the proof of Theorem 1 in [27]."
>
> ***Chain component*** is a standard concept  widely used in the literature for CPDAG (see, e.g., Andersson et al 1997; Castelletti et al. 2024). Chain components are the *maximal undirected connected subgraphs* in a CPDAG, and they are also referred to as *undirected and connected chordal graphs* (UCCGs).
>
> For any given CPDAG, its chain components are uniquely determined. Consequently, in Line 3 of the DSCS algorithm, the number $K$ of chain components is a constant for a given CPDAG structure. The value of $K$ depends on the CPDAG structure. In the extreme case where the CPDAG is a DAG, $K$ equals the number of vertices (each vertex forms its own chain component); when the CPDAG is fully undirected, $K=1$ (the entire graph is a single chain component). In practical examples, $K$ is typically small relative to the number of vertices.
>
> **Novelty and distinction**. Leveraging chain component properties for efficient verification of collapsibility is a novel contribution of our work, unique to the CPDAG setting. In DAGs, the idea of chain components does not apply, since there are no undirected edges; each node forms its own chain component only in the extreme case where the CPDAG reduces to a DAG. This structural distinction underlies both the theoretical and practical efficiency of our approach, as compared to previous methods that focus on DAGs alone [28].
>
> We thank the reviewer again for these suggestions and will clarify these points in the revised manuscript.
>
> **Reference**
>
> Andersson, S. A., Madigan, D. & Perlman, M. D. (1997). A characterization of Markov equivalence classes for acyclic digraphs. Ann. Statist. 25 505-5
>
> Castelletti, F., & Consonni, G. (2024). Bayesian sample size determination for causal discovery. Statistical Science, 39(2), 305-321.
>
>
> &nbsp;
>
> **Weakness 2**
> > Although the result of this work for CPDAGs are novel but given the previous definitions and characterisation of the c-removability, it is a limited contribution. In other words, it seems that given the previous results of estimation collapsibility in DAGs, its extension to CPDAGs is not trivial but a natural generalisation.
>
> **We appreciate the reviewer’s concern that our extension from DAGs to CPDAGs might appear "natural". In fact, as summarized in Section 1, our work overcomes several substantial and previously unaddressed challenges:**
>
> 1. **Introduce the sequential c‑simplicial set for CPDAGs.** The proposed *sequential $c$-simplicial set* required a careful and unified treatment of both directed and undirected subgraphs within a CPDAG, an extension that is not only non-trivial but also far from a natural generalization of prior work on DAGs. In particular, the connection with undirected graphs involves subtle and non-obvious technical challenges, which we address for the first time in this paper. See Appendix B for the technical details.
>
> 2. **Establish an order‑free characterization.** Establishing an order-free characterization of sequential $c$-simplicial (or sequential $c$-removable) sets for arbitrary node sets $M$  is remarkably challenging. To the best of our knowledge, even for DAG, no prior work has addressed this problem. A key difficulty here for CPDAG lies in  simultaneously accounting for both the complex internal structure of $M$ (which may contain a mixture of directed and undirected edges) and its intricate relationships with the surrounding graph structure. Our work makes significant headway on this problem by developing simple yet powerful equivalent characterizations (see Section 3.2).
>
> 3. **Exploit CPDAG structure with a divide‑and‑conquer strategy.** The theoretical insights directly led to the novel and efficient DSCS algorithm for verifying collapsible submodels.  By exploiting the unique structural properties of CPDAGs, our method enables a divide-and-conquer approach to collapsibility verification - processing each chain component independently. This strategy achieves substantial computational savings compared to conventional approaches that rely on sequential checking.
>
> **Collectively**,  these aspects constitute significant theoretical and algorithmic contributions that go well beyond a straightforward extension of earlier results. We believe these advances meaningfully enrich both the theory and practice of estimation collapsibility in the literature. We will ensure these points are clearly highlighted in the revised manuscript.

---

> > ### Comment · Reviewer_ufyg · 2025-08-05
> >
> > Thank you to the authors for their rebuttal. It addresses some of my concerns and I will update my score accordingly.

---

### Official Review · Reviewer_Ha3g · 2025-07-08

**Clarity:** 3
**Significance:** 4
**Originality:** 4
**Rating:** 4
**Confidence:** 3

**Summary:**

This paper tries to provide a unifying framework of estimation-collapsibility for CPDAG (completed partially directed acyclic graph), DAG (directed acyclic graph) and UG (undirected graph). Estimation-collapsibility refers to the ability to remove a set of variables from the graphs while preserving the structure of the rest. The authors achieve this by focusing on CPDAG, which is an Markov equivalent class of DAG, with connections to UG.

Following prior works, they give an equivalence relation between estimation-collapsibility and $c$-simplicial, a new notion proposed in this paper, under the non-trivial CPDAG assumption. On top of this equivalence result, this paper also gives a more efficient algorithm for detecting estimation-collapasibility via $c$-simplicial that improves the time complexity from $O(|V|^4)$ to $O(|V|^2)$.

**Questions:**

Question.
1. On Line 196, the authors mention $\textbf{ne}_{G^*}(v) = \emptyset$, I don’t see why this is true, unless you define the neighbors as the ones connected via *undirected* edges. There’s no formal definition of neighbors other than on Line 89 (which had no formal definition on the main text). To avoid confusion, I think it's best to make a note mentioning the distinction between neighbors v.s. parents and children -- that they are disjoint.
2. Does the $c$ in $c$-simplicial stands for conditional? I think it would be helpful to point out the meaning: easier to understand what's the difference between $c$-simplicial (CPDAG) and simplicial (undirected graph).
3. Why does the definition of $c$-simplicial (for CPDAG) seem so different from the counterpart on DAG? Intuitively, do we not expect that DAG would be reduced back to CPDAG (viewing DAG as a subset of CPDAG, or simply transforming DAG into its CPDAG)? I notice the discussion on Line 200-202: it only describes reduction from CPDAG to DAG but not the only way around. Maybe it is obvious to the authors, but I don't quite see how to convert $c$-removable to $c$-simplicial (worth writing a bit more on this). Are they incompatible?
   For example, consider the node $v$ in DAG: $v \rightarrow a \leftarrow b$. $v$'s $\mathbf{Mb}_{G}(v)$ is $a$ and $b$. They ($a$ and $b$) are adjacent and does not being to $\textbf{pa}_G(v)$. So it should be $c$-removable (and thus estimate-collapsible according to Proposition 1). But $v$ is not $c$-simplicial in CPDAG, since it is not a leaf node.
4. Since the definitions for CPDAG, DAG, UG and its related key concepts on $c$-simplicial, $c$-removable, simplicial are so spread out, I would suggest collecting and adding them to a table for easy comparison and reference.
5. What do you mean by "non-trivial", e.g., on Line 133? Perhaps should briefly explain it or put a footnote somewhere? Is there a definition for "chain component" which appears first on Line 263-264? I'm not entirely sure what it means in this context.
6. If I understand correctly, the DSCS algorithm doesn't actually provide a $O(|V|^2)$ upper bound for DAG? See the incompatibility doubt I raised in Question 3 (also see related discussion around Line 76-80).

Typos:
1. On line 245, "establishes CPDAG-based criterion for check estimation-collapsibility", "check"->"checking".

**Ethical Concerns:**

["NO or VERY MINOR ethics concerns only"]

**Final Justification:**

The authors have addressed my concerns adequately.

**Limitations:**

yes.

**Paper Formatting Concerns:**

No.

**Quality:**

4

**Strengths And Weaknesses:**

Strengths:
1. The problem is well-motivated: detecting estimation-collapasibility from CPDAG is a natural generalization of the DAG setting and it is interesting to see the connection from the UG perspective.
2. This paper is well-written, clear and mostly self-contained. I find the graphs are pretty helpful in understanding the paper.
3. Their algorithm is quite straight-forward given the equivalence theorem between estimation-collapasibility and $c$-simplicial.

Weakness:
1. I have some questions/concerns regarding applying $c$-simplicial criteria over the vanilla DAG setting -- see Question 3.
2. Some terminology is missing and might affect reading -- see Question 1 and 5.

Overall, I think this paper is solving a well-motivated problem that could be of interests to audience of this conference. Their techniques have a lot of connections to existing works, though I do have some doubts (perhaps due to my limited understanding).

---

> ### Author Rebuttal · Authors · 2025-07-29
>
> Thank you for your careful reading. Your valuable feedback helps us to improve the clarity of the manuscript. We will add more detailed explanations for the important concepts in the main text. We will also thoroughly check the grammar and spelling. Regarding the issues you raised, we address the questions in detail below.
>
> &nbsp;
>
> **Question 1**
> > On Line 196, the authors mention $\mathbf{ne}_{G^*}(v) = \emptyset$, I don't see why this is true, unless you define the neighbors as the ones connected via undirected edges. There's no formal definition of neighbors other than on Line 89 (which had no formal definition on the main text). To avoid confusion, I think it's best to make a note mentioning the distinction between neighbors v.s. parents and children -- that they are disjoint.
>
> We appreciate your attention to this subtle but important distinction. In our notation, "neighbors" indeed refer to those vertices connected via *undirected* edges, while "parents" and "children" are those via *directed* edges. We provided the definitions of "neighbors", "parents" and "children" in Appendix A (see the paragraph after Line 500). To improve clarity, we will include these important definitions in the main text in the final version of the manuscript.
>
> &nbsp;
>
> **Question 2**
> > Does the $c$ in $c$-simplicial stands for conditional? I think it would be helpful to point out the meaning: easier to understand what's the difference between $c$-simplicial (CPDAG) and simplicial (undirected graph).
>
> The "$c$" in $c$-simplicial stands for *compound*, as the $c$-simplicial property (CPDAG) combines the features of a simplicial set (undirected graph) and a leaf set (DAG). Thank you for point this out. We will clarify this in the manuscript.
>
> &nbsp;
>
> **Question 3**
> >Why does the definition of $c$-simplicial (for CPDAG) seem so different from the counterpart on DAG? Intuitively, do we not expect that DAG would be reduced back to CPDAG (viewing DAG as a subset of CPDAG, or simply transforming DAG into its CPDAG)? I notice the discussion on Line 200-202: it only describes reduction from CPDAG to DAG but not the only way around. Maybe it is obvious to the authors, but I don't quite see how to convert $c$-removable to $c$-simplicial (worth writing a bit more on this). Are they incompatible? For example, consider the node $v$ in DAG: $v \rightarrow a \leftarrow b$. $v$'s $\mathbf{Mb}_{G}(v)$ is $a$ and $b$. They ($a$ and $b$) are adjacent and does not being to $\textbf{pa}_G(v)$. So it should be $c$-removable (and thus estimate-collapsible according to Proposition 1). But $v$ is not $c$-simplicial in CPDAG, since it is not a leaf node.
>
> Thank you for your careful review. As established in Theorem 1 (Lines 223-238), $c$-removable and $c$-simplicial are compatible.
>
> The confusion may have arisen as our notation $\mathbf{Mb}\_{\vec{G}}(v)$ refers to the ***union of the vertex $v$ and its Markov boundary*** $\mathbf{mb}\_{\vec{G}}(v)$ (see Lines 88-91 and the paragraph after Line 502). That is,
> $\mathbf{Mb}\_{\vec{G}}(v)=\mathbf{mb}\_{\vec{G}}(v) \cup \\{v\\}$.  In your example, for the DAG $v\rightarrow a\leftarrow b$, $\mathbf{Mb}\_{\vec{G}}(v)$ consists of $a$, $b$, **as well as $v$**, where $b$ and $v$ are not adjacent. Therefore, $v$ is *not $c$-removable*, which is consistent with $v$ not being $c$-simplicial in the CPDAG.
>
> To enhance clarity, we will reiterate the definition of $\mathbf{Mb}\_{\vec{G}}(v)$ in Definition 2 in the revision.
>
> The intuition for the equivalence between $c$-removability for DAG and $c$-simpliciality for CPDAG is as follows. In the reference [28], $c$-removability is characterized via Markov equivalence classes: a vertex $v$ is $c$-removable from a DAG $\vec{G}$ if and only if there exists some DAG $\vec{G}^\prime$ which is Markov equivalent to $\vec{G}$ such that $\mathbf{ch}\_{\vec{G}^\prime}(v)=\emptyset$. Inspired by this fact, in the CPDAG $G^{\*}$, we require $\mathbf{ch}\_{G^{\*}}(v)=\emptyset$; otherwise, for every DAG $\vec{G}\in \mathcal{M}(G^{\*})$, $\mathbf{ch}\_{\vec{G}}(v)=\emptyset$ would fail. Moreover, we must constrain the neighbor set $\mathbf{ne}\_{G^{\*}}(v)$ so that for all $a,b\in \mathbf{ne}\_{G^{\*}}(v)$ (i.e., $a-v-b$), there exists some DAG $\vec{G}^\prime\in \mathcal{M}(G^{\*})$ with $\mathbf{ch}\_{\vec{G}^\prime}(v)=\emptyset$ and with edges $a\rightarrow v\leftarrow b$. Since this orientation cannot create a new $v$-structure, $a$ and $b$ must be adjacent in CPDAG $G^*$. These conditions naturally lead to the definition of a $c$-simplicial vertex in a CPDAG, and hence to the concept of a sequential $c$-simplicial set. Because $c$-removability and $c$-simpliciality are compatible, our DSCS algorithm offers an efficient way to determine $c$-removability.
>
> &nbsp;
>
> **Question 4**
> > Since the definitions for CPDAG, DAG, UG and its related key concepts on $c$-simplicial, $c$-removable, simplicial are so spread out, I would suggest collecting and adding them to a table for easy comparison and reference.
>
> Thank you for the helpful suggestion. We will include a table summarizing all key concepts. These additions, which will be placed in the appendix, will enhance the clarity and accessibility for a broader audience.
>
> &nbsp;
>
> **Question 5**
> > What do you mean by "non-trivial", e.g., on Line 133? Perhaps should briefly explain it or put a footnote somewhere? Is there a definition for "chain component" which appears first on Line 263-264? I'm not entirely sure what it means in this context.
>
> Thank you for your suggestion.
>
> As stated in Line 131,  ***non-triviality*** is a mild distributional constraints proposed in the work of [28].  It is pointed out in [28] and we quote here that: "the DAG models for both contingency tables and the Gaussian distributions are non-trivial," and furthermore, "non-triviality is satisfied by most DAG models and is also implicitly used in the proof of Theorem 1 in [27]."
>
> As for ***chain component***, this is a standard concept in the literature for CPDAG (see, e.g., Andersson et al 1997; Castelletti et al. 2024). Chain components are the *maximal undirected connected subgraphs* in a CPDAG, and are also referred to as *undirected and connected chordal graphs* (UCCGs).
>
> These clarifications will be incorporated in the revision.
>
> **Reference**
>
> Andersson, S. A., Madigan, D. & Perlman, M. D. (1997). A characterization of Markov equivalence classes for acyclic digraphs. Ann. Statist. 25 505-5
>
> Castelletti, F., & Consonni, G. (2024). Bayesian sample size determination for causal discovery. Statistical Science, 39(2), 305-321.
>
> &nbsp;
>
> **Question 6**
> > If I understand correctly, the DSCS algorithm doesn't actually provide a $O(|V|^2)$ upper bound for DAG? See the incompatibility doubt I raised in Question 3 (also see related discussion around Line 76-80).
>
> Thank you for raising this important complexity issue.  As in the response to Question 3,  $c$-removable and $c$-simplicial are compatible. Therefore, our proposed DSCS algorithm can be employed to check estimate collapsibility for DAGs.
>
> For a given DAG, our approach involves two steps: (1) converting the DAG to a CPDAG, which is $O(|V|^3)$ (Chickering 2002), and (2) applying our DSCS algorithm, which has worst-case complexity $O(|V|^2)$. Thus, the overall procedure is $O(|V|^3)$. Nevertheless, our method improves upon the naive $O(|V|^4)$ approach and, to our knowledge, represents the most efficient algorithm available for this problem.
>
> Currently, we are not aware of any algorithm that can check estimate collapsibility for general DAGs with $O(|V|^2)$ complexity. We will clarify this point and the open complexity question in the revised manuscript, and we believe our work lays the groundwork for future progress in this direction.
>
> **Reference**
>
>  Chickering D M. Optimal structure identification with greedy search[J]. Journal of Machine Learning Research, 2002, 3(Nov): 507-554.

---

> > ### Comment · Reviewer_Ha3g · 2025-08-05
> >
> > I would like to thank the authors for their clarifications and insights. I believe they have adequately addressed my concerns. I believe the proposed changes would improve the presentation of this paper. I will update my score accordingly.

---

### Note · Authors · 2025-08-13

We thank all reviewers for their constructive feedback and recognition of our work’s **novelty**, **efficiency**, and **practical relevance**. This paper presents the **first unified theory and algorithm** for estimation-collapsibility in CPDAGs, delivering **non-trivial theoretical advances** alongside **scalability to high-dimensional Bayesian networks**.

###  **Key Contributions**

- **Theoretical Breakthrough** — We introduce the **sequential $c$-simplicial set** for CPDAGs, unifying directed and undirected subgraphs and addressing technical challenges — a *novel and non-trivial result*.
- **Order-Free Characterization** — We establish the first order-free characterization of sequential $c$-simplicial ($c$-removable) sets for *any* node set, handling both internal and external connectivity in CPDAGs.
- **Algorithmic Innovation** — Our **DSCS algorithm** leverages CPDAG chain components in a divide-and-conquer approach, reducing complexity from $O(|V|^4)$ to $O(|V|^2)$. Following reviewer feedback, we will integrate maximum cardinality search for neighborhood-clique checks, reducing the complexity to linear within each chain component, achieving an overall complexity of $O(|V| + |E^*|)$.
- **Practical Relevance** — Our results address the realistic case where only Markov equivalence classes are available, making the method applicable to observational data.
- **Clarity and Validation** — The paper is structured for accessibility, with examples, experiments, and **new large-scale real-world evaluations**.

###  **Addressing Reviewer Concerns**

- **Self-Containment** — We will integrate key definitions (e.g., chain components) into the main text for clarity.
- **Beyond a “Natural” Extension** — We emphasize that our CPDAG extension goes far beyond DAG generalization, introducing sequential $c$-simplicial sets, the first order-free characterization for mixed-edge node sets, and a chain-component-based divide-and-conquer method for major computational savings.
- **Expanded Experiments** — We have now added results on three large real-world Bayesian networks, showing consistent and increasing advantages of DSCS over DCR. DCR [28] is the most efficient DAG baseline; our work is the first to address CPDAGs.

We have **fully addressed all major concerns**, with reviewers acknowledging these resolutions. We appreciate the constructive dialogue with the reviewers, which has helped strengthen both the theory and practical impact of this work.

---

### Decision · Program_Chairs · 2025-09-17

**Decision:**

Accept (poster)

**Comment:**

The paper proposes a more efficient technique for estimation collapsibility, which is very relevant when performing inference in high dimensions for a small subset of variables. The results for completed partially directed acyclic graphs generalizes prior results for directed acyclic graphs and undirected graphs. Experimental results validate the main claims.

The reviewers unanimously agree to accept this paper. For a camera-ready version, please take into the account the comments from all reviewers regarding for instance, defining neighbors/parents/children in the main text, including other concepts such as c-simplicial/c-removable/non-triviality/chain-components, and including additional benchmark network experiments.